# Training deep learning models with a multi-station approach and static aquifer attributes for groundwater level simulation: what's the best way to leverage regionalised information?

Sivarama Krishna Reddy Chidepudi[a][b], Nicolas Massei[a], Abderrahim Jardani [a], Bastien Dieppois [c], Abel Henriot [b], Matthieu Fournier [a]

[a] Univ Rouen Normandie, UNICAEN, CNRS, M2C UMR 6143, F-76000 Rouen, France
[b] BRGM, 3 av. C. Guillemin, 45060 Orleans Cedex 02, France
[c] Centre for Agroecology, Water and Resilience, Coventry University, Coventry, UK

Correspondence to : Sivarama Krishna Reddy Chidepudi
(sivaramakrishnareddy.chidepudi@univrouen.fr)

**Abstract.** In this study, we used deep learning models with advanced variants of recurrent neural networks, specifically Long Short-Term Memory (LSTM), Gated Recurrent Unit (GRU), and Bidirectional LSTM (BiLSTM), to simulate large-scale groundwater level (GWL) fluctuations in northern France. We developed a multi-station collective training for GWL simulations, using "dynamic variables (i.e., climatic) and static basin characteristics. This large-scale approach offers the possibility of incorporating dynamic and static features to cover more reservoir heterogeneities in the study area. Further, we investigated the performance of relevant feature extraction techniques such as clustering and wavelet transform decomposition with the aim of simplifying network learning using regionalised information. Several modelling performance tests were conducted.  Models specifically trained on different types of GWL, clustered based on the spectral properties of the data, performed significantly better than models trained on the whole dataset. Clustering-based modelling reduces complexity in the training data and targets relevant information more efficiently. Applying multi-station models without prior clustering can lead the models to learn the dominant  behaviour preferentially, ignoring unique local variations. In this respect, wavelet pre-processing was found to partially compensate clustering, bringing out common temporal and spectral characteristics shared by all available time series even when these characteristics are "hidden" because of too small amplitude. When employed along with prior clustering, thanks to its capability of capturing essential features across all time scales (high and low), wavelet decomposition used as a pre-processing technique provided significant improvement in model performance, particularly for GWLs dominated by low-frequency variations. This study advances our understanding of GWL simulation using deep learning, highlighting the importance of different model

training approaches, the potential of wavelet pre-processing, and the value of incorporating static attributes.

## 1. Introduction

Understanding the large-scale hydrological functioning of a hydrosystem is the best approach for grasping a more global view of water reserves and implementing appropriate long-term management strategies (Kingston et al., 2020; Massei et al., 2020). However, this approach requires constructing a large-scale hydrological model capable of capturing interactions over large areaswhile respecting hydraulic continuity across the hydrosystem. The model must be able to sanalyse and test, for example, the effects of different modes of exploitation or any other human interventions, as well as the effects of climate change over the long term. Building the large-scale model implies collecting and processing a massive database to accurately capture all the geological, oceanic, climatic, and anthropogenic forcings that drive groundwater flow.

However, the numerical, physics-based representation of all the physical processes occurring during the hydrological cycle in the subsurface remains an extremely complex task to achieve rigorously, particularly in large-scale modelling (Paniconi & Putti, 2015). Although progress has been made in this field, applications of physics-based models are still mainly focused on aquifers in relatively small watersheds.

Under these conditions, data-driven tools have emerged as an interesting alternative (or complement) for capturing the complex interactions that occur on different time and space scales, including large ones. They rely on efficiently processing a large database without having to rely on numerical physical representations of the non-linear physical processes that link climatic and hydraulic signals(Hauswirth et al., 2021). These processes are efficiently approximated on the basis of small and simple weight matrices defined to reproduce the observed hydraulic signals, either at an aquifer or river(Vu et al., 2023). The application of artificial intelligence (AI) algorithms, and deep learning (DL) in particular, is growing in the geosciences and especially in the hydrosciences(Nourani et al., 2014, 2023; Rajaee et al., 2019), thanks to the increase in computational resources, but also the growing availability of global datasets for different hydrological variables(Addor et al., 2017; Kratzert et al., 2023), which are making it possible to better address issues related to the understanding and management of hydrological systems (Muñoz-Carpena et al., 2023). This growing interest has been confirmed in several recent studies that have highlighted the potential of deep learning tools for hydrological simulations(Fang et al., 2022; Klotz et al., 2022; Kratzert et al., 2019, 2021; Nourani et al., 2021) and forecasting tasks (Jahangir et al., 2023; Momeneh & Nourani, 2022; Sina Jahangir & Quilty, 2023; Vu et al., 2023). Most often, these approaches are applied to rainfall-runoff modelling due to the availability of long-term runoff data, which is not always the case for aquifers due to the high cost of installing piezometers.

Furthermore, the highly heterogeneous nature of underground reservoirs leads to complex hydrodynamic behaviours on a regional or continental scale, which cannot be captured by a limited number of piezometers. Consequently, the few applications of DL to groundwater, whether in simulation (Chidepudi et al., 2023a) or forecasting (Bai & Tahmasebi, 2023a; Collados-lara et al., 2023; Rahman et al., 2020; Vu et al., 2023; Wunsch et al., 2021), are on a local scale and involve only single station models on a small number of piezometers in the construction of neural networks.

DL models have proved effective on a local scale and are also on a larger scale by collectively training a significant number of piezometers (Chidepudi et al., 2023b; Heudorfer et al., 2024). This collective approach involves using and processing all available piezometric stations to learn about relationships or events likely to occur at the target station, even if they have not yet been observed at that station. This approach also requires using and extracting the relevant global climate signal and tracking its effects. This can have a delayed effect on piezometric fluctuations, making DL models more effective for long-term forecasting.

Working with groundwater data also presents unique challenges compared to runoff data, such as 1) complex and heterogeneous geological factors influencing GWLs, 2) difficulty in linking the available data to the appropriate well (for surface water, this is easily done through catchment delineation, but this isn't the case for aquifer delineation), 3) slow response time (longer time series needed, i.e. data availability issue as mentioned above), 4) distinct sensitivities to human activities (e.g. pumping), which differ from those affecting runoff data, like river straightening and dam construction.

In some hydrological studies, the term 'global models' is being used to describe models trained from multiple wells or stations. However, this term can be misleading in the groundwater context as it suggests a broader scope than intended. Therefore, in this study, we use the term "multi-station approach" for models trained on data from different wells with external input variables, which more accurately reflects their scope and methodology.

Efforts to use data from multiple GWL stations in model training have been limited and have often focused on forecasting or reconstruction using data from nearby GWL wells as input. For example, Vu et al. (2021) used data from nearby stations to reconstruct the GWLs at a single station, albeit using GWLs from nearby stations only while training individual models for each station. Another recent study (Patra et al., 2023) developed so-called 'global models' for GWL forecasting and not simulations, i.e. these models only use past GWL data to forecast future GWLs. (Bai & Tahmasebi, 2023) used graph neural networks for GWL forecasting to capture the spatial dependencies of nearby wells and compared their performance with the single station gated recurrent unit (GRU) and long short-term memory (LSTM). A recent study by Gholizadeh et al. (2023) used LSTM alongside static attributes and demonstrated its applicability for simulating both streamflow discharge and GWL. However, the scope of the study for GWL simulations was limited to only two dynamic

variables: precipitation and temperature. This approach was used to simulate 21 GWL wells across Alabama from 1990 to 2021. Notably, the study focused on annually varying GWLs, which may not represent the most difficult GWL variations to model. Cai et al. (2021), in their study conducted in the central eastern continental United States, showed that GRU performed better when it was informed by hydrogeological characteristics expected to affect groundwater response along with dynamic input variables (in this case, precipitation and streamflow).

Several studies on groundwater modeling also demonstrated the potential of clustering methods (Nourani et al., 2022) in hybrid models along with AI approaches such as self-organising map (Nourani et al., 2015, 2016; Wunsch et al., 2022b), K-means (Ahmadi et al., 2022; Kardan Moghaddam et al., 2021; Kayhomayoon et al., 2021, 2022; Nourani et al., 2023), Fuzzy  C-means ((Jafari et al., 2021; Nourani & Komasi, 2013; Rajaee et al., 2019; Zare & Koch, 2018). However, most of these studies mainly focused on autoregressive approaches that rely on using previous GWL or nearby wells' GWL data as input for forecasting or reconstruction.

The regionalisation of GWLs, a process that could involve clustering and training of DL models using the non-autoregressive approach of learning from external input variables on comprehensive datasets, remains underexplored. The potential of multi-station approaches, particularly those that integrate static attributes and dynamic data or use clustering/pre-clustering, remains largely unevaluated in the context of GWL simulations. While these methods have proven effective in runoff modelling(Fang et al., 2022; Hashemi et al., 2022; Klotz et al., 2022),  their application to GWL simulation is still not fully explored or validated across diverse hydrogeological settings. A comprehensive evaluation of their strengths and weaknesses is essential to unlock their full potential in the simulation of GWLs. This includes a detailed investigation of the performance of these models in various GWL simulation scenarios. In addition, techniques such as wavelet pre-processing, such as BC-MODWT (Chidepudi et al., 2023a), have shown promise in single-station models but have not been extensively tested on regional-scale simulations. Given this background, the current study aims to address several research questions:

a) How do the generalised (multi-station) models compare with the specialised (single-station) models in simulating GWLs?

b) Can wavelet pre-processing techniques improve the performance of models for different types of GWLs when trained with data from all available stations?

c) To what extent do static attributes or one-hot encoding techniques help models  generalise across different GWL behaviours? Is using a combination of these methods more effective than using them individually? Furthermore, how do these models compare to those trained on GWL stations grouped by similar spectral and temporal statistical characteristics?

130    d) What are the key variables that influence the learning of these models, particularly in terms of capturing low-frequency variability while it is buried into high-frequency-dominated explanatory signals?

By addressing these questions, this study aims to provide a comprehensive evaluation of regional modelling approaches for GWL simulations and to compare their performance with the local approaches. We want to

135 highlight that the present study is not dedicated to 'forecasting' as is the case in most applications of DL to groundwater modelling. The reader can be referred to Beven & Young (2013) for distinctions between 'simulation' and 'forecasting'. In brief, according to their framework, 'simulation' means reproducing system behaviour without using observed outputs, while 'forecasting' involves reproducing system behaviour ahead of time based on past observations. This study focuses on simulation to understand GWL dynamics rather

than forecasting future levels. This distinction is important for framing our approach and interpreting our results. To achieve this, we test different approaches for multi-station models while including static attributes and comparing the results with those obtained using local models. Furthermore, we evaluate the impact and usefulness of integrating wavelet pre-processing with multi-station deep learning models. All our experiments are conducted only under the gauged scenario, similar to (Li et al., 2022)

The rest of the paper is structured as follows: Section 2 details the datasets used, and Section 3 presents the methodology and experimental design for the different approaches. Section 4 discusses the ability of the models and robustness in capturing different variations in GWLs and input scenarios. Section 5 deals with the discussion on the interpretability of the obtained results. Section 6 presents our main conclusions and perspectives.

## 2. Study area and Data

The study area is approximately 80,000 km2 of Northern France, as depicted in Figure 1. The available GWLs of climate-sensitive wells (i.e. not strongly affected by human activities and sensitive to climate variability (Baulon et al., 2022a)) with high data quality until the end of 2022 were obtained from the ADES (Accès aux Données sur les Eaux Souterraines) database (https://ades.eaufrance.fr/; Winckel et al., 2022). All the wells

considered in the study are in unconfined aquifers. In addition, the GWL data were clustered into three different clusters following the methodology outlined by Baulon et al. (2022b), which is based on spectral properties (i.e. characteristic time scales of variability inherent to each cluster). These clusters are identified as annual, mixed, and inertial, as depicted in Figure 1. Specifically, the first cluster showcased in Figure 1 exhibits a pattern predominantly influenced by the annual cycle, indicating an annual behaviour. The second

cluster, the mixed, shows characteristics of both annual and interannual variability. The third cluster, the inertial, is mainly characterised by its low-frequency variability, as shown in Figure 1. The dataset consists of 35 mixed, 23 inertial and 18 annual stations. All the wells considered in the study are in unconfined aquifers.

A comprehensive list of all analysed wells, including their identifiers, GWL type and coordinates, is available in the supplement (Table S1).

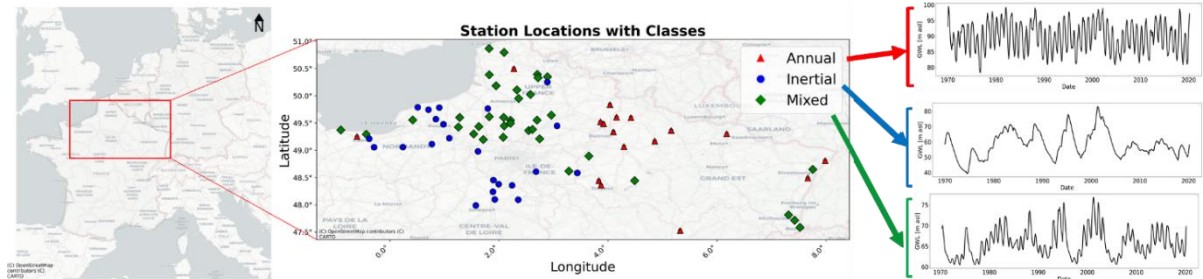

Figure 1: Clustering of GWL timeseries data (Background layer: © OpenStreetMap contributors 2023. Distributed under the Open Data Commons Open Database License (ODbL) v1.0.) based on the spectral statistical properties (Baulon et al., 2022b)

We used the forcing data from ERA5 (Hersbach et al., 2020) with a spatial resolution of 0.25 degrees to obtain the dynamic climate variables. In particular, we extract seven atmospheric variables: 10m zonal (W-E) U-wind component (u10), 10m meridional (S-N) V-wind component (v10), 2m air temperature (t2m), evaporation (e), mean sea level pressure (msl), surface net solar radiation (ssr), total precipitation (tp). These variables are among the most commonly used inputs for hydrological and land surface models, representing atmospheric conditions and circulation, moisture fluxes and radiative forcing. ERA5 is the best available global reanalysis with the data available from 1940 and is generally considered adequate for capturing regional and global hydrometeorological variations. Addressing the uncertainty issue of ERA5 is beyond the scope of this paper and can be considered a complete research work. ERA5 Reanalysis data have uncertainty related to potential regional biases; this and their use for hydrological modelling is still ongoing research, particularly in "large-sample hydrology"  (Maria Clerc-Schwarzenbach et al., 2024.). Precipitation is considered to have more bias than temperature. However, recent studies conducted recently concluded that ERA5 temperature and precipitation biases had been consistently reduced compared to ERA-Interim and were found to be quite accurate for hydrological modelling, for instance, in the case of conterminous US (Tarek et al., 2020). Gualtieri (2022) highlighted that ERA5 uncertainties were greater in mountainous and particularly in coastal locations located less than 15 km from the coastline (in the study presented herein, only 1 station out of 76 is located within the 10-15 km range identified in Gualtieri (2022)). Finally, one recent study (Lavers et al., 2022) conducted by ECMWF on evaluating ERA5 precipitation for climate monitoring concluded that using ERA5 precipitation should be recommended for extra-tropical regions. However, for our study area, we have been evaluating different potential alternative reanalysis products, such as the SAFRAN (Système d'Analyse Fournissant des Renseignements Atmosphériques à la Neige) reanalysis developed specifically for France (Vidal et al., 2010). ERA 5 and SAFRAN precipitation appeared to have the

190 same low-frequency components as detected in the GWL time series, as displayed in Figure.3 (this paper) and Fig.11 in Chidepudi et al. 2023a. ERA 5, then, is suitable for our purpose.

Static attributes are available for different ranges of aquifer classes with different resolutions; we took the static attribute's value corresponding to each well's location—associated with the Well IDs. Static attributes, coming from the BDLISA (Base de Donnée des Limites des Systèmes Aquifères) (https://bdlisa.eaufrance.fr/)
database, are point-scale information, i.e., each well received set of attributes given different possible methods (geographical imputation, rule-based, human expertise). BDLISA is based on a mix of information from geological maps, piezometric maps, and hydrochemistry at a scale of 25km. BDLISA was originally designed at a 25km scale and later upscaled to larger scales. For our study, we kept information coming from BDLISA at its original scale (25km), which means aquifer static attributes have a resolution of 25km. This
information from BDLISA should be understood as a local-to-regional description of aquifers.

In this work, we also included static attributes (Table 1 and Figure 2) to assess whether such informative data would help to better represent small differences between GWL time series owing to different contexts (e.g., type of porosity, overall geological context, lithology, location (lon, lat)). Such data were retrieved from the French national database BDLISA ); they would be related to the filtering capabilities of the aquifers with
205 respect to the input signals (e.g. precipitation). Although they seem somehow redundant, they are expected to provide complimentary information about the hydrogeological nature of the hydrosystems. Exact details of static attributes for each GWL station can be found in the supplement (Table S1).

Table 1: Summary of the static attributes used in the current study. Comprehensive explanation of all descriptions can be found at the URLs provided in the 3rd column.

| Variable | Description | Possible values and details |
|---|---|---|
| type of porosity | Type of environment for a hydrogeological entity characterised based on the level of porosity: porous, karstic, fracture.... | https://id.eaufrance.fr/nsa/353 |
| geological context at large-scale | Hydrogeological entity theme based on the different geological formations: alluvial, sedimentary, volcanic... | https://id.eaufrance.fr/nsa/348 |
| lithology | Dominant rock types associated with the well location: limestone, clay... | https://id.eaufrance.fr/nsa/165 |
| co-ordinates | latitude and longitude of the well location | |

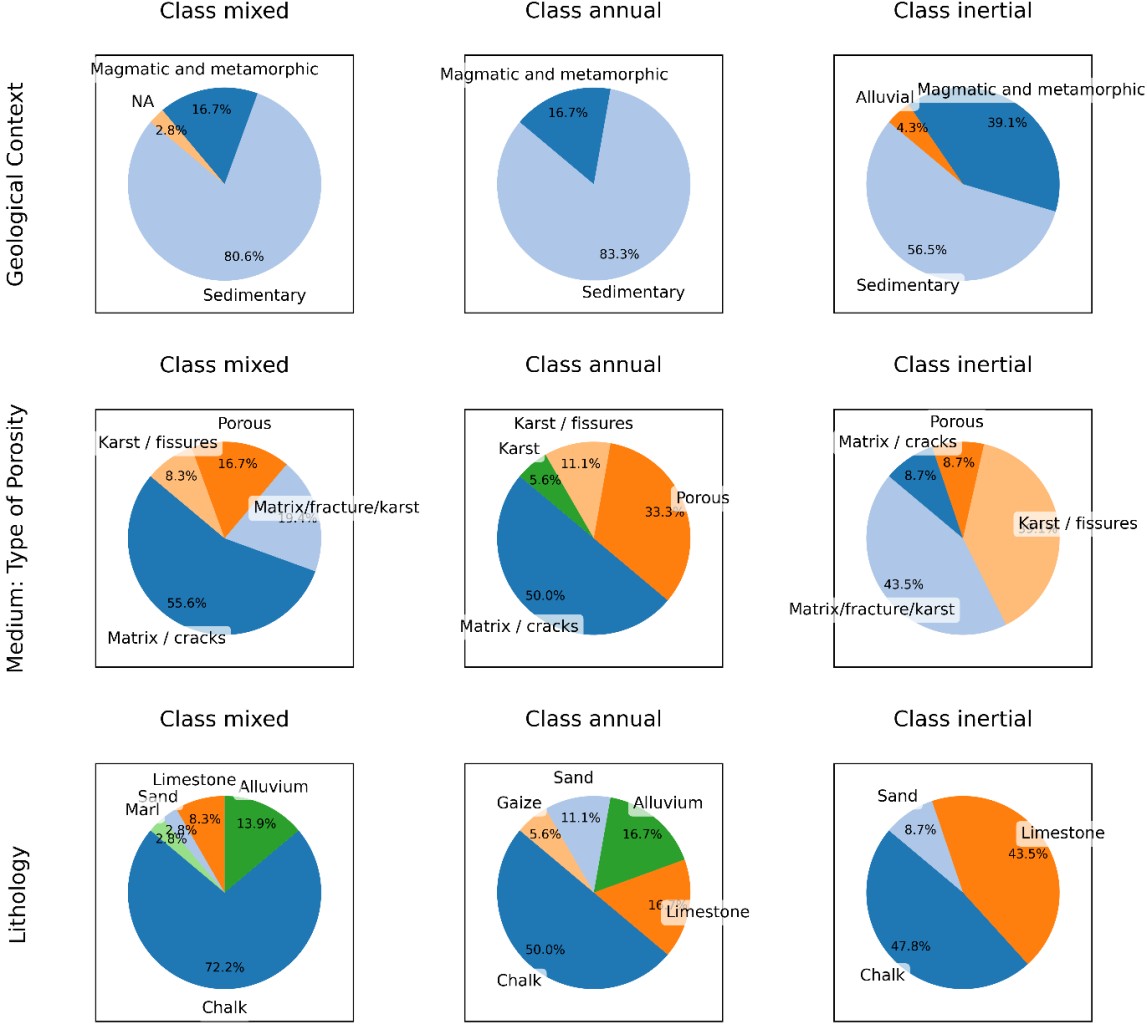

Figure 2: Distribution of Geological Features by Class

The decision to include the relevant static attributes comes from a trade-off between the transposability of models and the availability of attributes, as we have to make sure that all those variables are widely available at required resolution. Also, for some attributes like hydraulic conductivity, it might not be straightforward to get the most relevant resolution, which is needed to account for the most appropriate characteristic describing the well. For instance, a 25km resolution might not be relevant when aquifers are highly heterogeneous. Exploring the role of static attributes in more detail would require much further work than what was conducted in this study.

# 3. Methodology: from single station to multi-station training

## 3.1 Theoretical modelling background

In the current study, we explored the use of recurrent-based deep learning models to simulate GWLs across multiple stations using different approaches as described in section 3.2. We apply three types of recurrent neural networks: Long Short-Term Memory (LSTM, Hochreiter & Schmidhuber, 1997), Gated Recurrent Unit (GRU, Cho et al., 2014), and Bidirectional LSTM (BiLSTM, Graves & Schmidhuber, 2005), alongside a wavelet pre-processing strategy (BC-MODWT). Each of these methods is designed to process data that changes over time, capturing patterns and dependencies that occur over extended periods. In brief, LSTM has a single memory cell and three gates (forget, input, and output) to manage the flow of information. GRU simplifies this design, with only two gates (reset and update), to increase computational efficiency by reducing the number of parameters compared to LSTM. BiLSTM further optimises data analysis by simultaneously processing sequences in both forward and backward directions. These models are particularly good at identifying various patterns in data sequences, making them ideal for simulating GWLs that change over time (Vu et al., 2023).

We also explored the potential of wavelet decomposition (BC-MODWT) to decompose the data into components of varying frequencies (Figure 3), from high to low, to provide more detailed input to the DL models to better simulate the GWLs. As explained in Chidepudi et al. (2023a), decomposition depth (i.e. the choice of the number of components) was constrained by the trade-off between 1- achieving a sufficient high level of decomposition to ensure the low-frequency variability is properly reached, and 2- keeping the number of coefficients affected by boundary conditions as low as possible since these have to be ultimately removed from the input time series. All input time series were decomposed using BC-MODWT, with decomposition depth of 4 as in Chidepudi et al. (2023a). Figure 3 illustrates the decomposition result for the precipitation time series. A 4-level decomposition efficiently extracted the first 4 so-called wavelet details (tp_1 to tp_4) while the last fifth (so-called smooth) tp_5 component remains of sufficiently low frequency. It is clearly visible that tp_5, almost invisible in the original tp precipitation time series, corresponds well to the variability of the most interial GWL types (Figure.3, in red, with a few month time lag with respect to tp).

## 3.1.1 Model training and evaluation

To maintain consistent comparison criteria across all methods evaluated in the study, Bayesian optimisation was used for hyperparameter tuning. Details of range of hyperparameters used are shown in Table 1. Furthermore, the range of hyperparameters used for optimisation was standardised across all methods, following the best practices outlined for both standalone and wavelet-assisted models, as detailed in Chidepudi et al. (2023a) and Quilty and Adamowski (2018).

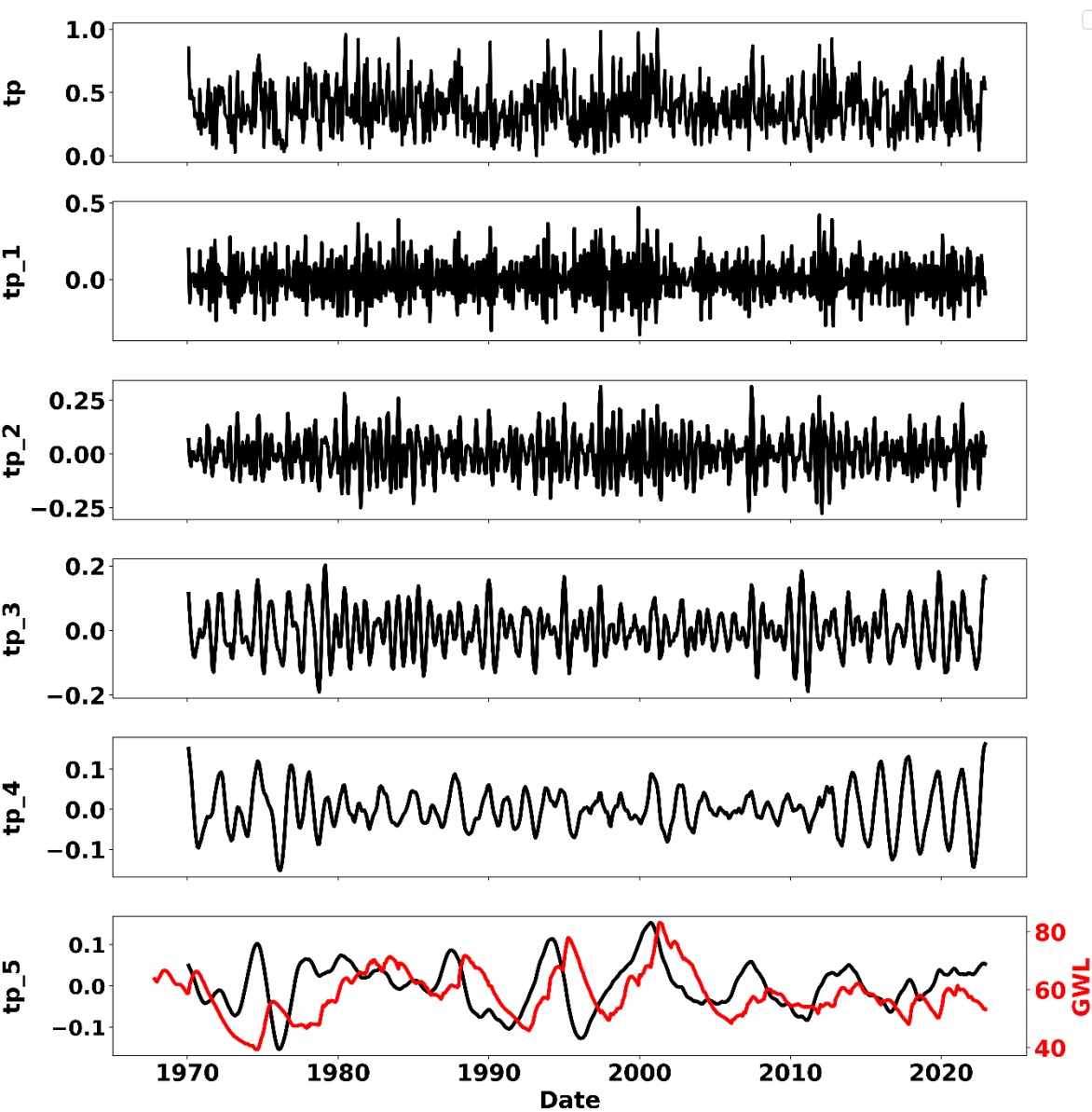

Figure 3: Total precipitation(tp) and its wavelet components: High(tp_1) to low frequency(tp_5) and GWL (in red).

However, we made an important update to the model architecture by setting the number of layers to one for all models, rather than optimising it. This decision was based on findings (Figure 4) that optimising the number of layers did not significantly improve performance and was in line with recent studies in related fields like rainfall-runoff modelling (Kratzert et al., 2019, 2021). Other adjustments included reducing the number of initialisations to 10 and setting the number of trials in the Bayesian optimisation to 30. These changes were aimed at reducing the computational requirements of our approach, making it more efficient without significantly affecting the quality of our results and are consistent with recent studies (Wunsch et al., 2022a).

Table 2: Hyperparameter details (Modified and adapted from chidepudi et.,al 2023a)

| Hyperparameter | Value considered |
| --- | --- |
| Sequence length | 48 |
| Dropout | 0.2 |
| Optimizer | ADAM |
| Early stopping | 50 |
| Number of layers | 1 |
| Hidden neurons | (10, 20, …,100) by 10 |
| Learning rate | (0.001,0.01) (log values) |
| Batch size | (16, 32, …,256) by powers of 2 |
| Epoch | (50, 100, …,500) |

The intricacies and specific technical details of the architectures these models are well documented in the existing body of deep learning research applied to hydrological simulations, as detailed in several studies (Chidepudi et al., 2023a;2024; Fang et al., 2022; Kratzert et al., 2021; Li et al., 2022; Vu et al., 2023).

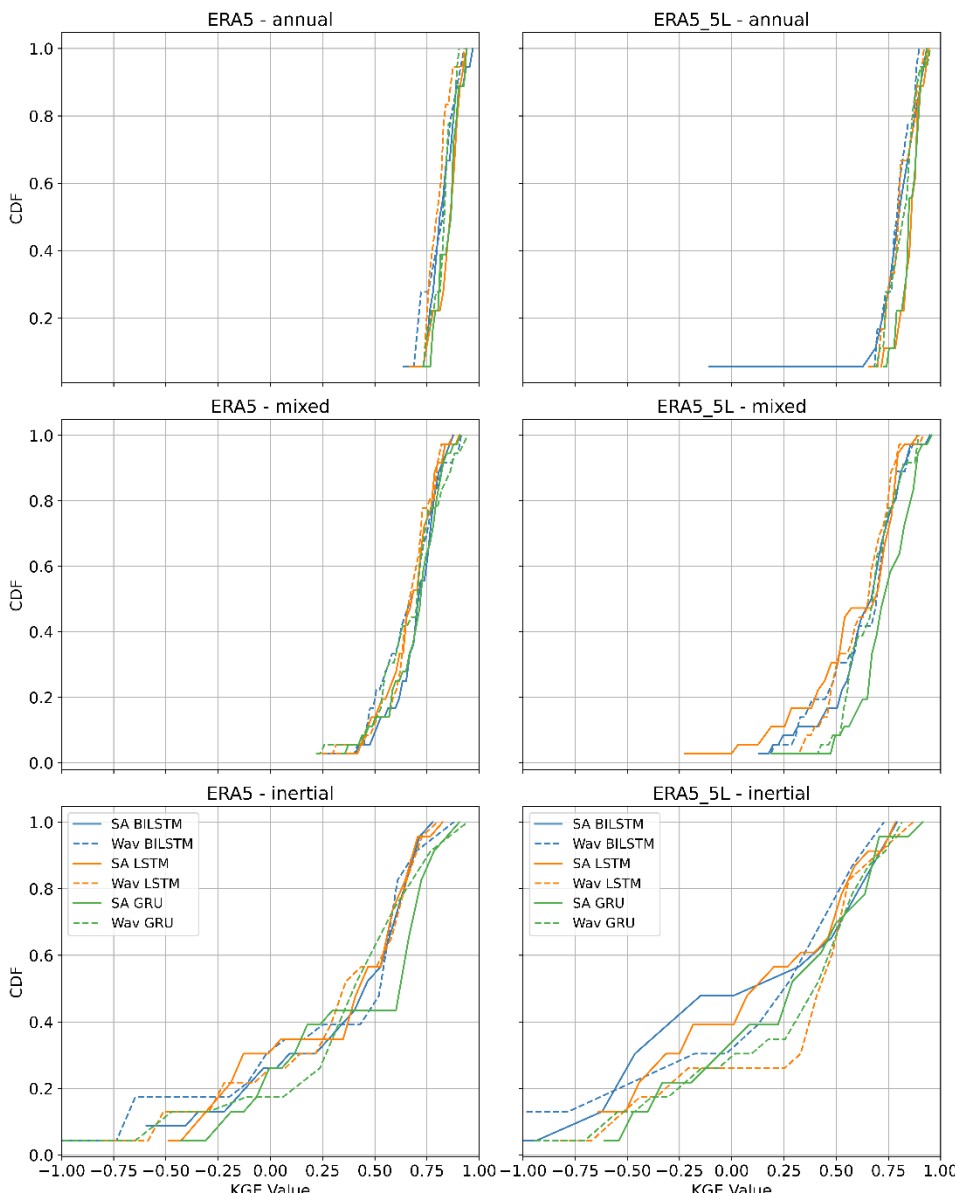

Figure 4: Comparison of performance of single layer DL models (left column) and multiple-layer DL models (right column) with respect to single station model as a reference. SA represents Standalone models while Wav represents Wavelet-assisted models.

To further interpret and decrypt the results for better understanding, we used the SHAP or Shapley Additive Explanations, approach(Lundberg & Lee, 2017), which is an increasingly popular game-centric approach for explaining the outcomes of deep learning models. SHAP, explains how each input feature influences the 'model's simulations. It does this by highlighting two key aspects: the importance of each variable, where a higher mean absolute SHAP value indicates a greater impact, and the nature of that impact, whether positive or negative.

## 3.2 Experimental design

This section details the experimental design used to assess the effectiveness of training models using data from all available stations. Our study uses different strategies to incorporate numerical and categorical data into the models. The aim is to improve the accuracy of GWL simulations by exploring ways of incorporating regional variability into the models. The experimental setup is structured to test different modelling strategies, as described below and visualised in Figure 5 & 6:

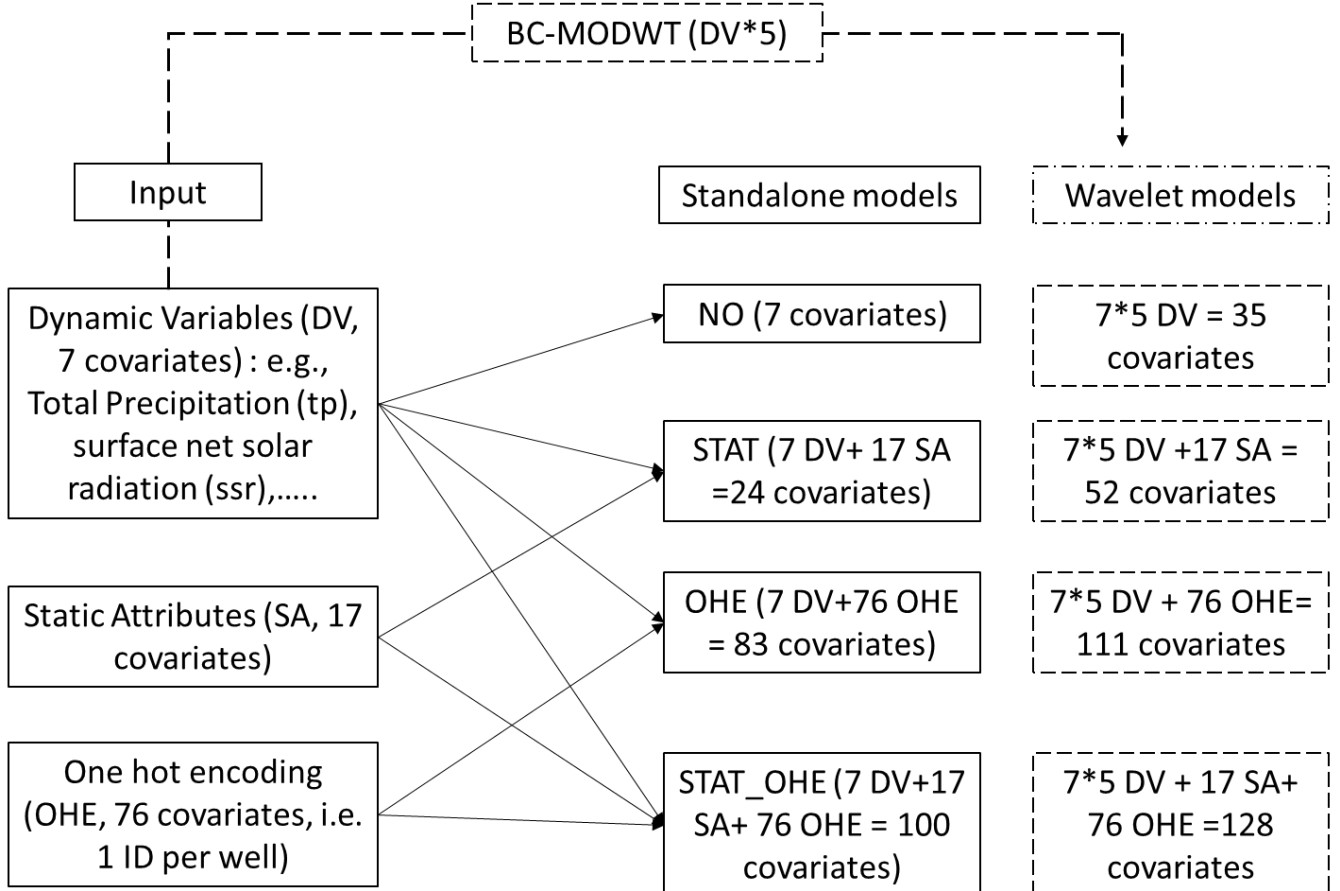

Figure 5: Construction of the different multi-station approaches for standalone and wavelet models and associated covariates (input features).

1. **Single station** or local models (models trained and tested individually per station): These models are trained and evaluated on data from individual stations. As a baseline, their performance provides a benchmark for evaluating the effectiveness of more generalised models. This approach is dominant in developing data-driven models for GWL simulations and is discussed in detail in Chidepudi et al. (2023a; 2024). The optimal hyperparameters for all standalone and wavelet models in the single-station approach are presented in the supplement (Table S3-S4).

2. **Multi-station** (models trained and tested together on many stations): These models are trained using data aggregated from multiple stations and tested with different input configurations.The covariates and input shapes for various multi-station approaches are summarized in Figure 5 and exact shape of 3D tensors are provided in supplement (Table S5):

   a. **NO (dynamic inputs only):** Models are trained on all stations using dynamic variables only, excluding static attributes and one-hot encoding.

   b. **OHE (One-Hot Encoding):** This method involves one-hot encoding to represent individual station ID information as binary vectors to ensure that the specific information is obtained from collective training, similar to the one-hot vector strategy developed in rainfall-runoff modelling (Li et al., 2022). This study showed that one-hot vector (one hot encoding using basin ID) could produce similar results to using catchment attributes in gauged basin scenarios. One-hot encoding serves as an alternative to incorporating static attributes directly into the model (Table 3).

Table 3: Example of one hot encoding based on different wells

| WELL | Dynamic variables | Well_ID_1 | Well_ID_2 | Well_ID_3 |
|------|-------------------|-----------|-----------|-----------|
| 1 | … | 1 | 0 | 0 |
| 2 | … | 0 | 1 | 0 |
| 3 | … | 0 | 0 | 1 |

   c. **STAT (Static attributes and dynamic Variables):** Models include both static attributes (e.g., latitude, longitude) and dynamic variables as inputs, with categorical variables encoded similarly to one-hot encoding but represented in separate columns for each unique value or class (Table 4).


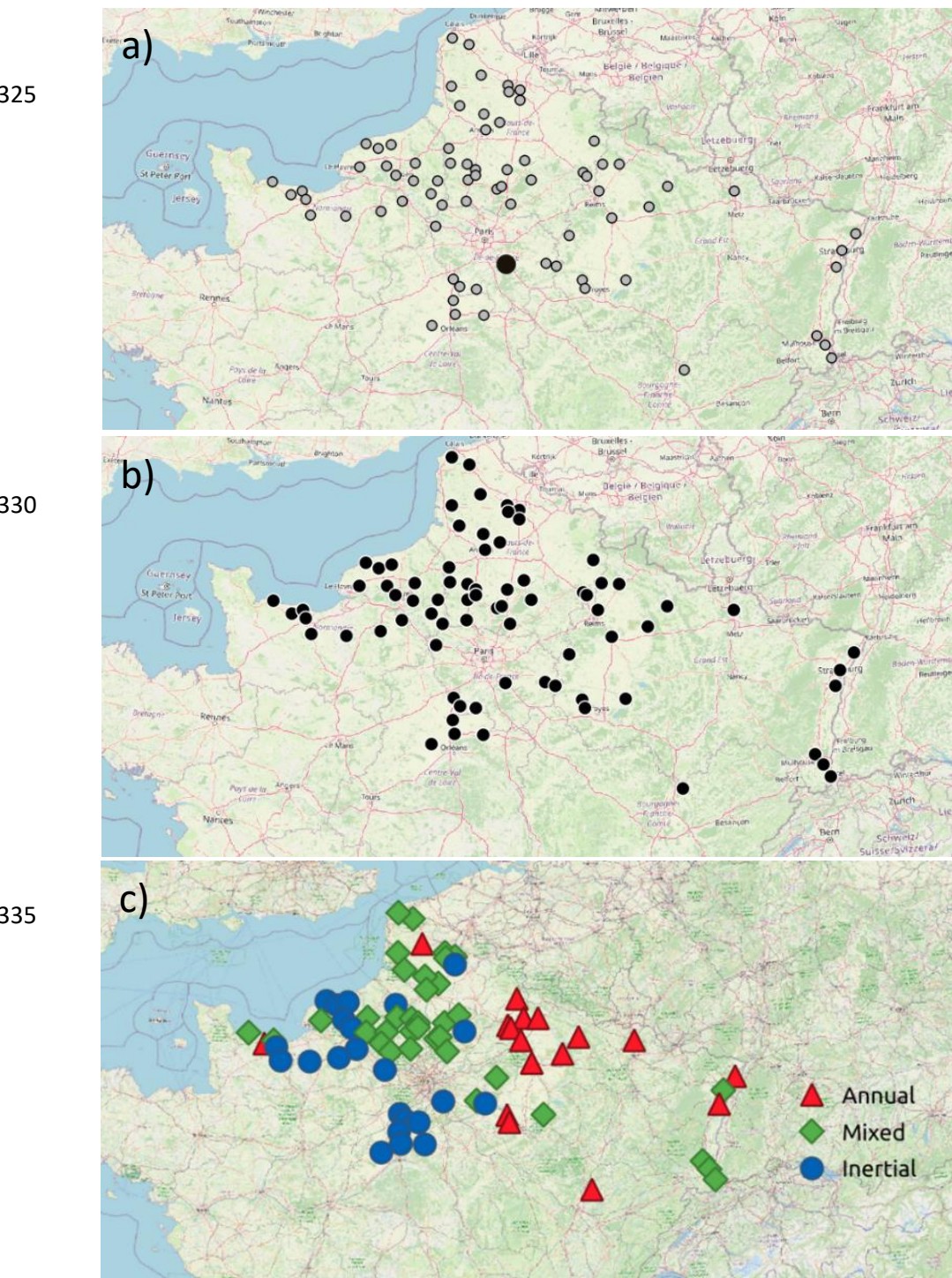



Figure 6: Comparison of different approaches adopted in the current study: a) single station (Top), b) multi-station without clustering (Middle) c) multi-station with clustering based on spectral properties(bottom). (Background layer: © OpenStreetMap contributors 2023. Distributed under the Open Data Commons Open Database License (ODbL) v1.0.)

3. **STAT_OHE (Static attributes, one-hot encoding, and dynamic variables):** This configuration
combines static attributes, one-hot encoding for well IDs, and dynamic variables to provide a

comprehensive dataset for model training. In other words, it is a combination of the two input strategies above.

Table 4: Example with static attributes of numeric and categorical types

| WELL | Dynamic variables | Static_1 (Lattitude) | Static_2 (Longitude) | Category_ 1 (Alluvial) | Category 2 (sedimentary) | Category 3 (Mountainous) |
|---|---|---|---|---|---|---|
| 1 | … | 5.1 | 9.5 | 1 | 0 | 0 |
| 2 | … | 2.8 | 10.8 | 0 | 1 | 0 |
| 3 | …. | 5.4 | 9.2 | 0 | 0 | 1 |

In addition to these configurations, we investigated the performance of multi-station models trained on GWLs with similar spectral statistical properties. This approach assesses the effectiveness of models tailored to specific GWL behaviours compared to more generalised models using the aforementioned strategies. In this study, Kling-Gupta efficiency (KGE, Gupta et al. 2009 ) is preferred over Nash–Sutcliffe efficiency (NSE) and other metrics because it offers a more comprehensive evaluation by integrating three aspects of model 355   error: correlation, bias, and the ratio of standard deviations.

For the single-station approach, the data was split into training (80%) and testing sets (20%) as described in Chidepudi et al., 2023. Furthermore, to facilitate hyperparameter tuning, the last 20% of the training data was used as a validation set. For the multi-station approach, the train-test split was also performed at each station, following the same procedure as the single-station approach. However, the data from all stations 360   was then collectively combined during the training. The rationale behind the specific train-test split is to ensure that the models capture the multi-annual to decadal variability in GWLs observed in the region. To achieve this, a minimum of 34 years of data (1970-2014) was used for training, while the most recent 8.66 years of data (2015/01-2023/08) were reserved for testing. This split corresponds to approximately 80% of the data for training and 20% for testing. By following this approach, we aimed to ensure that the models 365   were exposed to a sufficiently long period of data during training, enabling them to capture the amplitude and variability of GWL fluctuations over multi-annual to decadal timescales. The testing period was chosen to be the most recent years, allowing for an evaluation of the models' performance on the latest available data. The specific dates and periods used for training and testing at each station are detailed in the supplement (Table S2).

Our methodology for comparing single station and multi-station approaches, both with and without prior clustering based on spectral properties, is consistent with the research conducted in rainfall-runoff modelling by Hashemi et al. (2022), where the catchments were divided into five subsets according to hydrological regimes. This comprehensive experimental design aims to identify the most effective strategies for using

multi-station data in the simulation of groundwater level variations. Detailed hyperparameters for all the
multi-station standalone and wavelet models can be found in the supplement (Tables S6-S9)

# 4. Capabilities, performances and interpretability of multi-station approaches

## 4.1 Different strategies for multi-station approach

All models tested in the case of this study performed more or less equivalently and eventually yielded very satisfactory results. This can be attested by the performance comparison shown in Figure 4 (comparison of the 3 model types in single-station mode) and by comparing Figure 7 (GRU Multi-station) with Figures A1 (LSTM Multi-station)  and A2 (BiLSTM Multi-station). We finally decided to favor the GRU architecture owing to its recognised computational efficiency over more traditional LSTM-based architectures (Cho et al., 2014;
Cai et al., 2021; Chidepudi et al., 2023, 2024 )

Figure 7 shows the results of different GRU model configurations for simulating GWLs. The first row shows the performance of the standalone GRU model for different GWL categories, while the second row shows the wavelet-assisted GRU results.

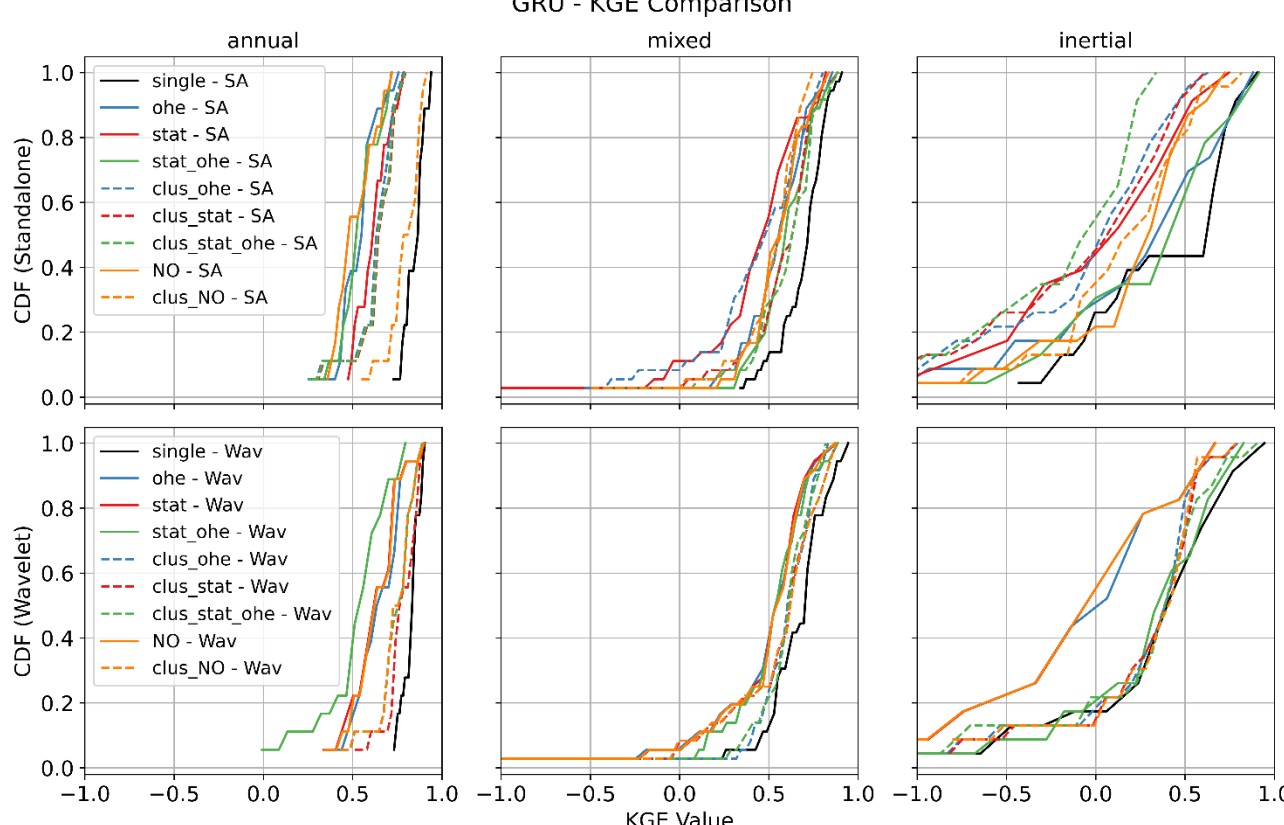

Figure 7: CDF Comparison of KGE values of the GRU With different approaches and GWL types.

Several observations can be made from Figure 7. Wavelet pre-processing generally improves model performance, especially in the inertial GWL category, where cumulative distribution functions (CDFs) are steeper and shifted to the right, indicating a higher proportion of simulations with high performance. This is in line with previous findings as already reported in our previous works (Chidepudi et al., 2023a & 2024). This demonstrates the wavelet decomposition ability to extract"hidden" inertial dynamics features which facilitates their assimilation by the model in the learning process. In other words, the improvement attributed to wavelet pre-processing becomes more pronounced as we move from annual to mixed, and then further to inertial behaviour. This is because in the case of annual-type GWL, the dominant variability (annual cycle) is already well expressed in several input variables (e.g. t2m, msl, ssr). In the case of mixed- and inertial GWL types, the dominant low-frequency variability, while also present, is barely expressed, almost "hidden", in the input data, and becomes prominent in GWL due to the low-pass filtering action of aquifers(Baulon et al., 2022; Schuite et al., 2019). Wavelet decomposition allows unraveling such hidden information, helping the neural networks to reach it for enhanced learning. This is illustrated in figure 3 with low-frequency component of precipitation (tp5) matching the variations of one intertial-type GWL (in red, with a few month-lag time), whereas it is masked by other higher-frequency components in the original precipitation time series (tp).The combination of static attributes and OHE gives competitive results,

particularly in the inertial category, demonstrating the effectiveness of this method without the need for prior clustering of GWL behaviour. Multi-station models, when trained separately for each GWL cluster, generally outperform those trained on aggregated data. This is reflected in higher KGE values for cluster-specific models, suggesting a better representation of the unique characteristics of each GWL type. However, this advantage diminishes for mixed GWLs, which are the majority in the study area. Although single station models perform best for all GWL types, some multi-station models approach or match their performance, highlighting their potential for regional-scale GWL simulations. For the annual GWL category, models trained on mixed GWL data without wavelet pre-processing and relying solely on static attributes do not show significant performance improvements, suggesting that static features alone may not adequately represent the dynamic nature of groundwater behaviour.

Figures 8-10 show the best GWL simulations obtained of different types (annual, mixed and inertial) for single and multi-station models. While single station models perform best, multi-station models are valuable where single station modelling is impractical either due to data limitations or computational requirements.

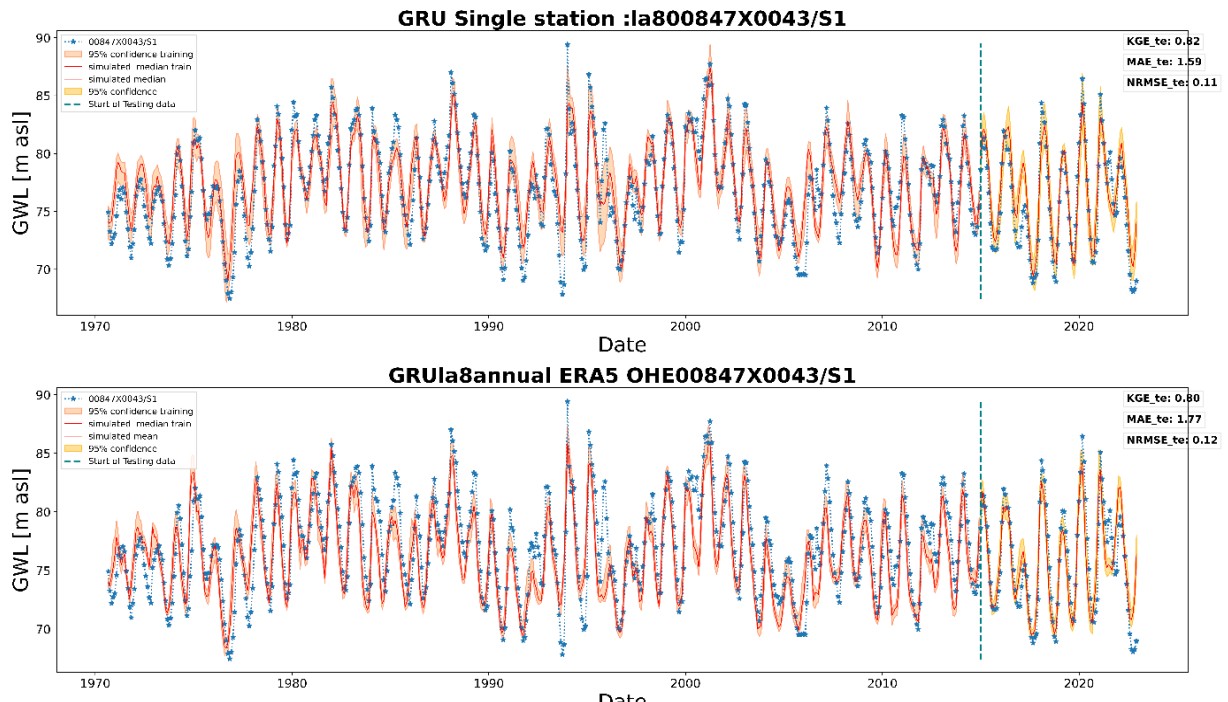

Figure 8: Results with wavelet assisted GRU in the annual type of GWLs through a) Single station (top) and b) Multi-station model trained on the annual type of GWLs with static and OHE (bottom)

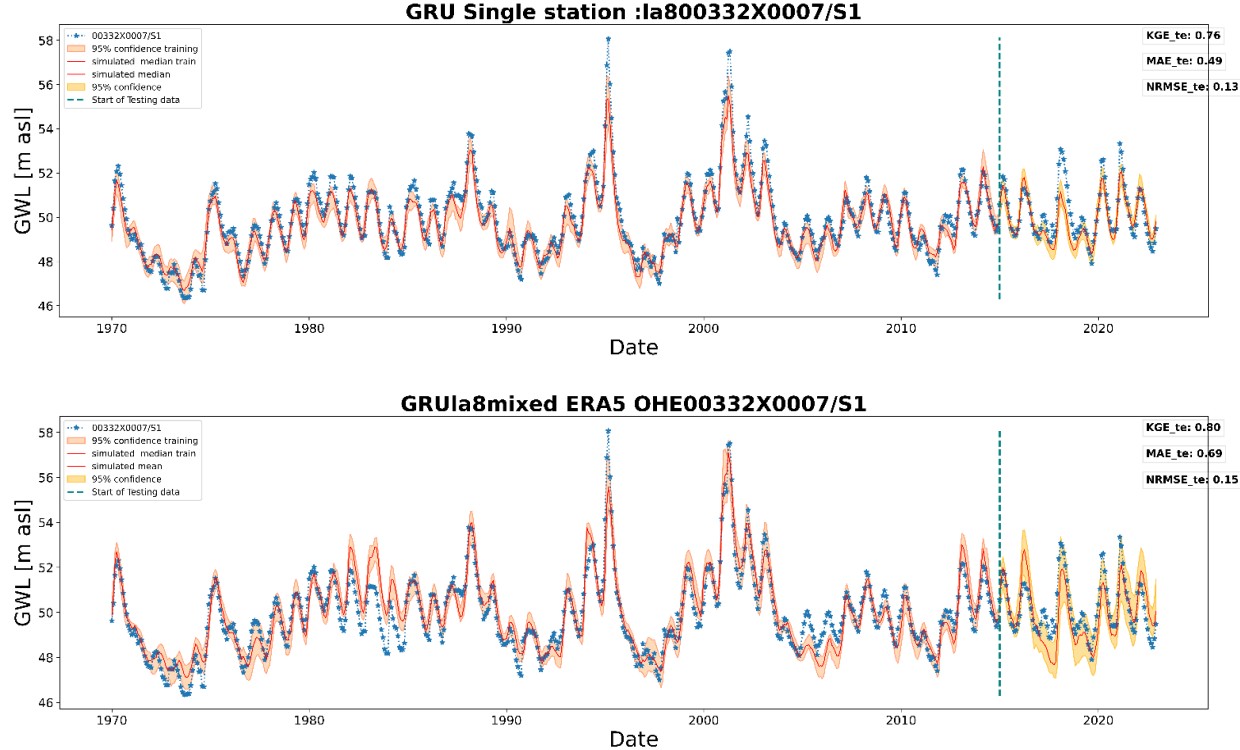

Figure 9: Results with wavelet assisted GRU in mixed type of GWLs through a) Single station (top) and b) Multi-station model trained

on the mixed type of GWLs with static and OHE (bottom)

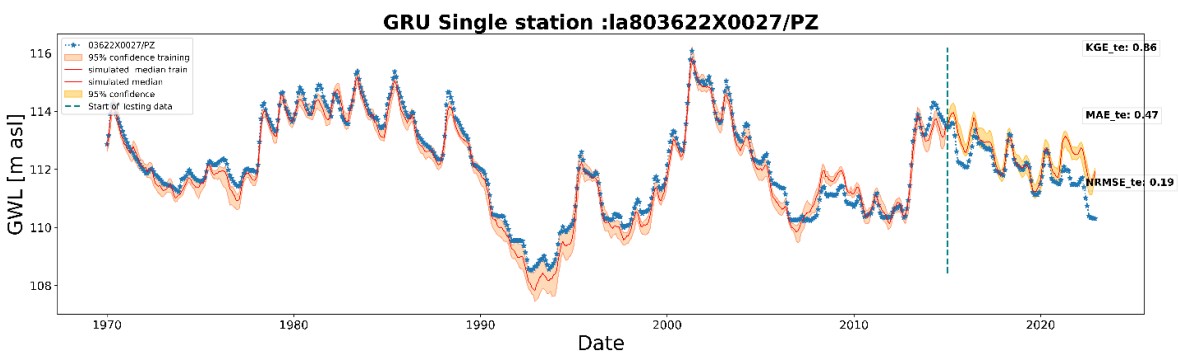

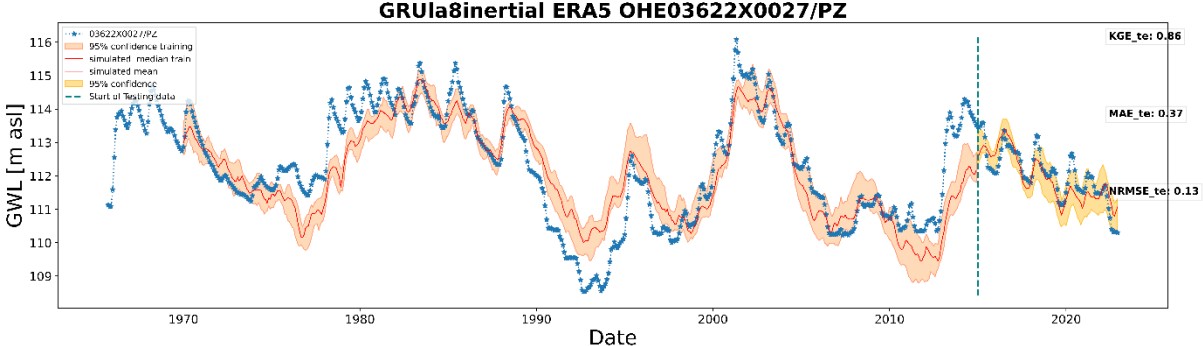

Figure 10: Results with wavelet assisted GRU in the inertial type of GWLs through a) Single station (top) and b) Multi-station model trained inertial type of GWLs with static and OHE (bottom)

In summary, wavelet-assisted GRU models are particularly effective, especially for low-frequency dominated GWL behaviour, and multi-station models designed for specific GWL types (i.e. training over specific pre-clustered datasets) generally outperform generalised models. The multi-station approach is sensitive to the dominant GWL type in the training dataset, with the best results seen in models trained for the predominant mixed GWL type in the study region To address the issue of model learning dominant behaviour in collective training of multi-station approaches, possible future investigation may involve generating synthetic time series with randomised amplitude changes of constituting frequencies to increase the dataset while balancing all the important behaviours. This could also help in understanding the influence of the size of dataset on using multi-station approaches.

## 4.2 Understanding GWL Simulations Through SHAP Interpretability

This section deals with the deeper understanding of the simulations from the insights obtained from the SHAP analysis on model's interpretability. In this study, we investigated the key contributing factors for GWL simulations in different approaches that were previously evaluated above in terms of accuracy.

Figure 11a shows the SHAP representative summary plot for the standalone models using a single station approach. These plots highlight the influence of different variables/attributes on the final simulation. In particular, the distribution of data points in the SHAP summary plots (Figure 11), with more points to the right (coloured red) indicating positive influences, and the opposite indicating negative relationships.

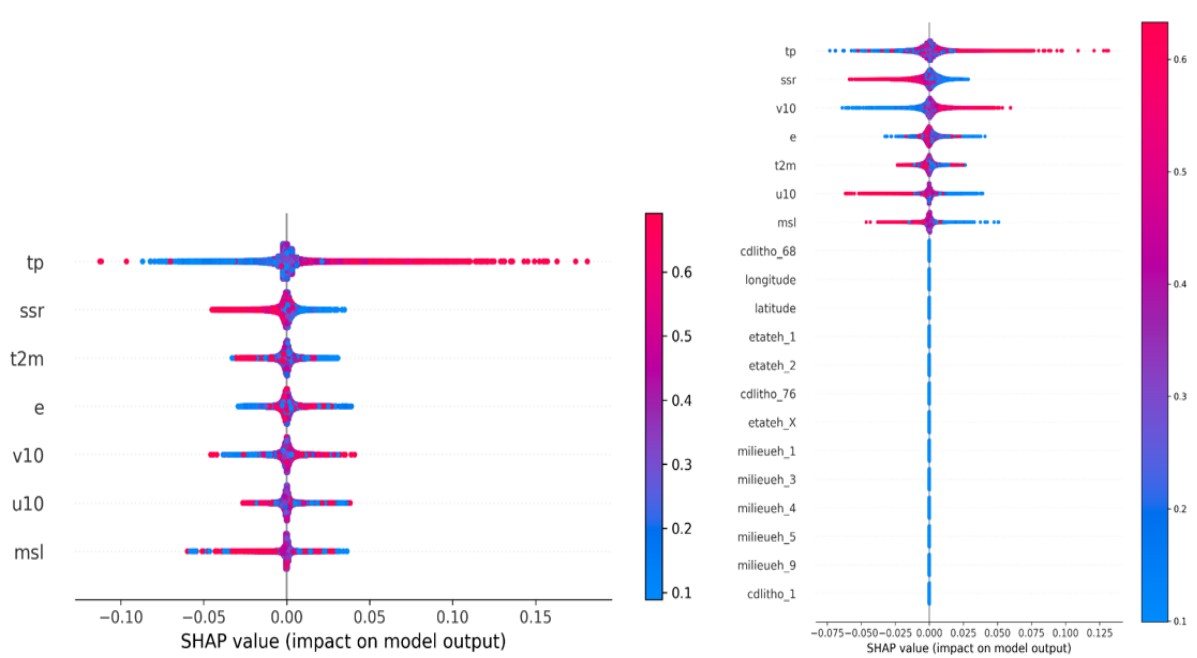

Figure 11: SHAP summary plot examples for single station model and multi station model with static attributes

From the analysis of Figure 12 and Figure 13, several notable patterns emerge regarding the contribution of different variables to GWL simulations using standalone models and those with wavelet pre-processing, and the impact of clustering as well as pre-clustering based on spectral statistical properties.

In single station standalone models, SHAP analysis shows that certain variables consistently influence GWL simulations, although their order of importance can change. Total Precipitation (TP) emerges as the key

factor, with Surface Net Solar Radiation (SSR) occasionally overtaking in mixed GWL cluster, especially in models trained on clusters, along with static features, or one-hot encoding (OHE). Nonetheless, TP and SSR are the primary drivers in these simulations.

In multi-station standalone models without clustering, TP and SSR lead in importance, followed by wind speed at 10 meters (v10), evaporation (e), and air temperature close to the ground (2-meter temperature,

t2m), which vary in their influence. Notably, v10 plays a bigger role in models in multi-station approaches. When models are trained on clusters, evaporation becomes more significant, yet the impact of clustering on variable importance is generally minor.

The spectral statistical characteristics (amplitude of high and low frequencies) were used for the pre-clustering of GWLs. These characteristics are related the filtering of the input signal by the physical properties

of the hydrological system. This highlights the importance of pre-clustering in capturing the physical characteristics of basins and suggests that it may be preferable to cluster based on these properties rather than relying on static attributes, especially when the relevance of static attributes is uncertain.

SHAP analyses show that standalone models maintain similar variable importance rankings even after clustering with static attributes and OHE. However, wavelet pre-processing shifts the importance towards

dynamic components, reducing the contributions of static features or OHE. When clustering is combined with wavelet pre-processing, low-frequency precipitation components emerge as key contributors, improving model performance.

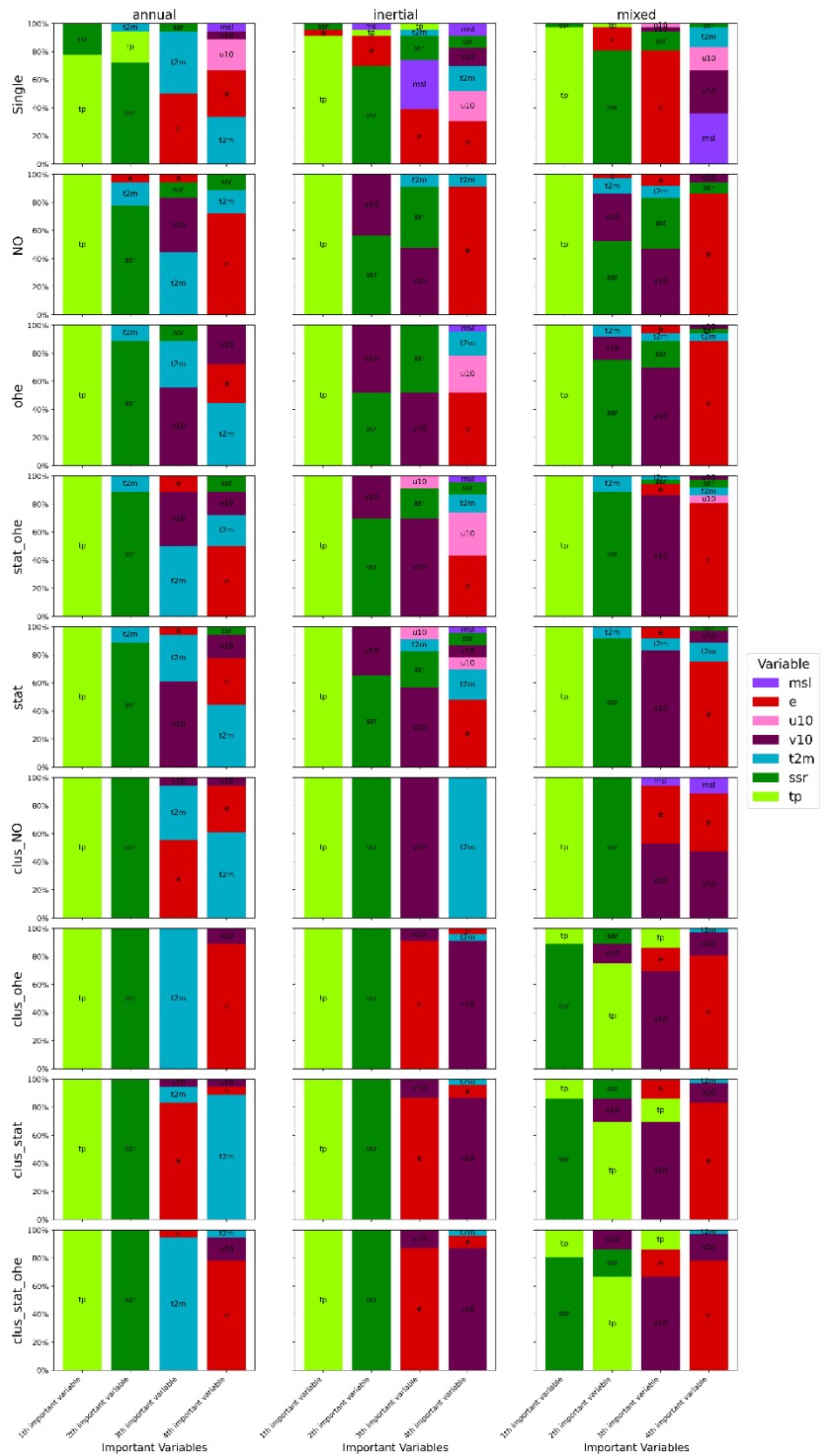

Figure 12: Top four important variables by cluster for standalone GRU models with different approaches. On Y-axis, Percentage of stations for each variable within in the cluster.

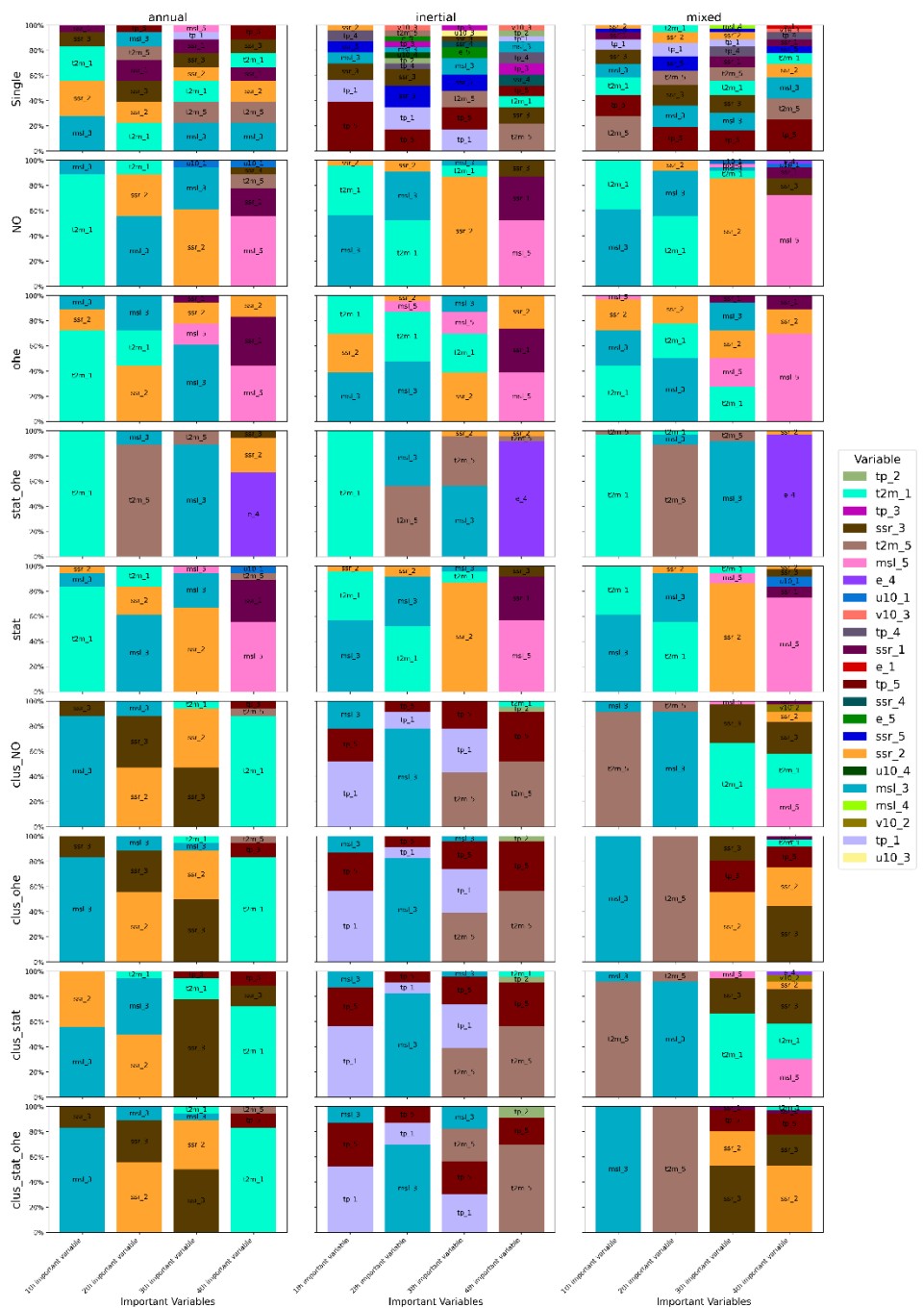

Figure 13: Top four important variables in regional GRU wavelet assisted model trained with different approaches for different classes

When models are trained after clustering, low-frequency components (e.g. , tp_5, t2m_5) are prioritised in mixed and inertial clusters: components not seen without clustering. Annual types prioritise relevant frequencies (1 to 3), consistent with single-station model patterns. The addition of static attributes to the OHE does not significantly alter the contributions, suggesting a dominance of dynamic variables after

decomposition. Also, differences among multi-station approaches after clustering are minimal for both

standalone and wavelet models.

Wavelet pre-processing performs a similar function to pre-clustering based on spectral properties by revealing information across all frequencies, including low amplitude frequencies that may be obscured. The order of best approaches based on the results: wavelet plus pre-clustering, followed by pre-clustering only, then wavelet only, and finally standalone highlighting the effectiveness of this approach.

There is a clear pattern when clustering is applied; without clustering the high frequency component of the 2-meter temperature (T2m_1) is dominant. Multi-station models show less diversity in variable contributions than single-station models. The exception is the Stat_OHE without clustering approach, which uniquely captures low-frequency information from T2m_5 and e_4. Otherwise, the static and NO approaches gave similar results.

The influence of static attributes or OHE appears to be minimal, possibly due to the high dimensionality introduced by numerous dynamic and static attributes. This observation suggests that future research could investigate alternative methods, such as target encoding, to address this dimensionality issue.

The purpose of the study presented here was not to determine the forcing factors of GWL variations; in this aim, a more comprehensive evaluation of such links would require specific approaches that have been
undertaken and presented in several previous works (Lee et al., 2019; Heudorfer et al., 2019; Liesch & Wunsch, 2019; Haaf et al., 2020; Giese et al., 2020). In some of our previous works (albeit for the Normandy region only), the linkages between GWL variability and potential forcing factors such as the thickness and lithology of surficial formations, aquifer thickness, vadose zone thickness, upstream/downstream location along the flow path, distance to the river, presence of karst,  were investigated using dedicated approaches
combining multivariate analysis, clustering and time series / spectral analysis and decomposition (Slimani et al., 2009; El Janyani et al., 2012 and 2014), which showed that GWL dynamics could be related to some basin and aquifer properties, although these relationships remained rather complex.  In a recent study, Haaf et al. (2023) developed an innovative methodological approach for modelling GWL at unmonitored locations using basin properties and machine learning on a daily time-step basis for alluvial aquifers with probably quite high
hydraulic conductivity overall. The models developed performed quite well in representing GWL variations at both intra- and interannual time scales using physiographic, land cover and geological characteristics. However, the amplitude of low-frequency, interannual to decadal variability of the dataset used in their study was much lower than what could be encountered in our monthly time step database. The specific type of aquifer Haaf et al. (2023) investigated likely explains their high sensitivity to many surface processes. In
our study, alluvial aquifers only represented approximately 10% of the GWL stations (8 over 76 stations) and were only of annual (3 stations)  or mixed (4 stations) types. Almost all other wells were located in chalk or

limestones. In the framework of our study, we decided to exclude some relevant characteristics such as vadose or saturated zone thickness: even when averaged over quite long periods (several years), these values actually represent GWL (the target variable). For mixed or inertial types in particular, it would probably make our models irrelevant for simulations over long-term predictions of several years or even decades when used along with climate projections and in another recent and relevant study by Heudorfer et al. (2024) developed entity-aware deep learning models with static attributes such as aquifer type (based on porosity). These authors concluded that the models did not show any entity awareness and eventually utilized static attributes as simple identifiers (almost similar to the OHE approach presented herein), meaning that the models did not make use of the relevant (hydro)geological information.

Although the added value of static variables was found to be marginal in the current study, they may prove useful in settings where no measurement is available. Further research is required to determine their utility in simulating such ungauged hydro systems. The approaches presented (except OHE) may be applicable to ungauged aquifers but require validation in a pseudo-ungauged environment. The use of data from multiple stations can enrich the dataset, improving the representation of groundwater systems and the robustness of the models. This multi-station approach also allows the model to be applied to areas without GWL monitoring, thereby capturing regional dynamics. However, single-station modelling remains important for understanding local interactions. The choice of method should therefore be guided by research objectives, data availability and the hydrogeological context. Where clustering results in too many groups, future studies should consider fine-tuning the general model for each cluster, following the approach of Mohammed & Corzo (2024).

## 5. Concluding remarks

This study has demonstrated the different multi-station approaches to GWL simulations with emphasis on the use of static attributes, one-hot encoding and the combination of both while training on all available data or by training on each GWL type based on the clustering. The study also highlights the potential of these approaches compared to the traditional single- station approach with and without the use of BC-MODWT. Key findings from this research highlight the advantages of clustering based on spectral properties, which have been shown to significantly improve the results of multi-station models, surpassing those of general models. Clustering is preferred over the use of static attributes, as the use of static attributes alone may not be sufficient to effectively distinguish different behaviours. Wavelet pre-processing is very effective at extracting relevant information at all levels, from high to low-frequency, allowing low-frequency dominated GWLs to be handled with increased accuracy. The combination of clustering and wavelet pre-processing produced the most accurate simulations, indicating that wavelet pre-processing likely captured key information needed for accurate modelling.

The study also showed that a multi-station approach, without clustering, should be used cautiously, as models tend to adopt dominant behaviour, which may not always be desirable. In scenarios where wavelet pre-processing is not applied, the combination of static attributes and OHE demonstrated promising results, particularly for GWLs dominated by low-frequencies. However, the minimal effect of static attributes or OHE observed in wavelet-assisted models may be due to the high-dimensional nature of these variables (due to 555 wavelet decomposition that increases the number of covariates), suggesting a potential avenue for future research to explore alternative encoding strategies, such as target encoding. SHAP analyses consistently identified key contributors across models, with clustered models highlighting the pivotal role of low-frequency components, especially precipitation and temperature, in achieving superior simulations for inertial and mixed types of GWL.

In this article, we introduced the following question: "What's the best way to leverage regionalised information?". In light of our results, it then seems like this is highly dependent on the amount and types of static attributes. It is generally expected that a much higher number of static attribute types would allow for a much better improvement of the multi-station simulation approach. However, Heudorfer et al. (2024) found no improvements using around 28 static features (including 18 environmental and ten time series-565 based). Also, as pointed out by these authors, employing static attributes for model training might be more relevant in applications on larger scales and/or larger datasets.

Moreover, one must remember that a trade-off must be found between the amount of static attributes required and data availability, especially for applications at ungaged sites. However, the use of static attributes and OHE yielded similar results in the gauged scenario (this study) and proved efficient in 570    accounting for local station information, which aligns with the findings of Heudorfer et al. (2024). On the other hand, in the study presented herein, wavelet pre-processing allowed for deciphering the "hidden" dynamic components of GWL variability (i.e. by separating low-frequency variations from annual or intra-annual variability), which eventually corresponded to taking into account the influence of (hydro)geological, geomorphological and physiographic properties. Ultimately, the latter – which varies across the study region 575    - operates a differential filtering effect of the input signals. Pre-clustering the dataset also yielded significant improvements that were even more noticeable when combined with wavelet pre-processing. However, owing to its capability of leveraging pre-processing the different frequency components in the time series of the whole dataset, wavelet pre-processing somehow acts in the same way as pre-clustering, which consists of grouping inertial (i.e. low-frequency dominated), mixed and annual time series in different clusters.

In summary, although the study has led to a better understanding of GWL simulation approaches with limited static attributes, further research is needed in the following areas, also exploring other physical basin parameters such as the thickness of overlying formations, altitude, distance from the sea, etc. It should also

be pointed out that clustering can be a source of information on the physical properties of the basin. Indeed, the three groups determined in this study on the basis of spectral properties indirectly carry information on the modalities of water transfer in the shallow formations and aquifer, which are controlled by the hydraulic properties of the basin. The study of the importance of using static data in groundwater modelling using deep learning tools needs to be extended to cover level prediction at sites with no piezometers. The insights gained here pave the way for future efforts to simulate GWLs in unmonitored or new locations, taking advantage of the robustness offered by multi-station models while recognising the value of single-station models for capturing local-scale interactions. Finally, it is noticeable through our study that the overall approach is compatible with a frugal AI approach (keeping in mind that our datasets are very small compared to classsical big datasets from other fields like natural language processing etc.): compact networks were tested and preferred (one layer), Bayesian optimisation was used instead of grid search for hyperparameter tuning. In addition, multi-station approaches pave the way for transfer learning, reducing the need for specialised models and retraining new models. The way forward is clear: to improve the GWL simulations efficiently, we may need to adopt a nuanced mix of efficient input signal pre-processing, potentially new encoding strategies to incorporate all possible additional knowledge of the system, and possibly clustering. Yet, we would recommend using advanced pre-processing over clustering, which would allow for leveraging the same type of information while preventing from separating the dataset and reducing its size.

Competing interests. The contact author has declared that none of the authors has any competing interests

## **Data availability and Acknowledgement**

We acknowledge the computational resources provided by CRIANN to carry out the experiments carried out as part of our ongoing project. All this work was conducted in Python version 3.8.13, and DL models were built using TensorFlow ((Abadi et al., 2016)) and Keras ((Chollet, 2015)). All figures were prepared using Matplotlib (Hunter, 2007), pandas (McKinney, 2010), and NumPy (Harris et al., 2020). Bayesian optimization was performed using the Optuna software (Akiba et al., 2019). All background maps in figures are from OpenStreetMap.

CRediT authorship contribution statement

**Sivarama Krishna Reddy Chidepudi:** Data curation; Formal analysis; Writing and conceptualization of original draft; model development and model runs; Investigation;

**Nicolas Massei:** Funding acquisition; Supervision; Writing and co-conceptualisation of the original draft- review & editing; Project administration.

**Abderrahim Jardani:** Supervision; writing, review & editing; Project administration.

**Bastien Dieppois:** review & editing;

**Abel Henriot :** Supervision; review & editing; Project administration.

**Matthieu Fournier**: review & editing


# Appendix A:

Results from LSTM and BILSTM

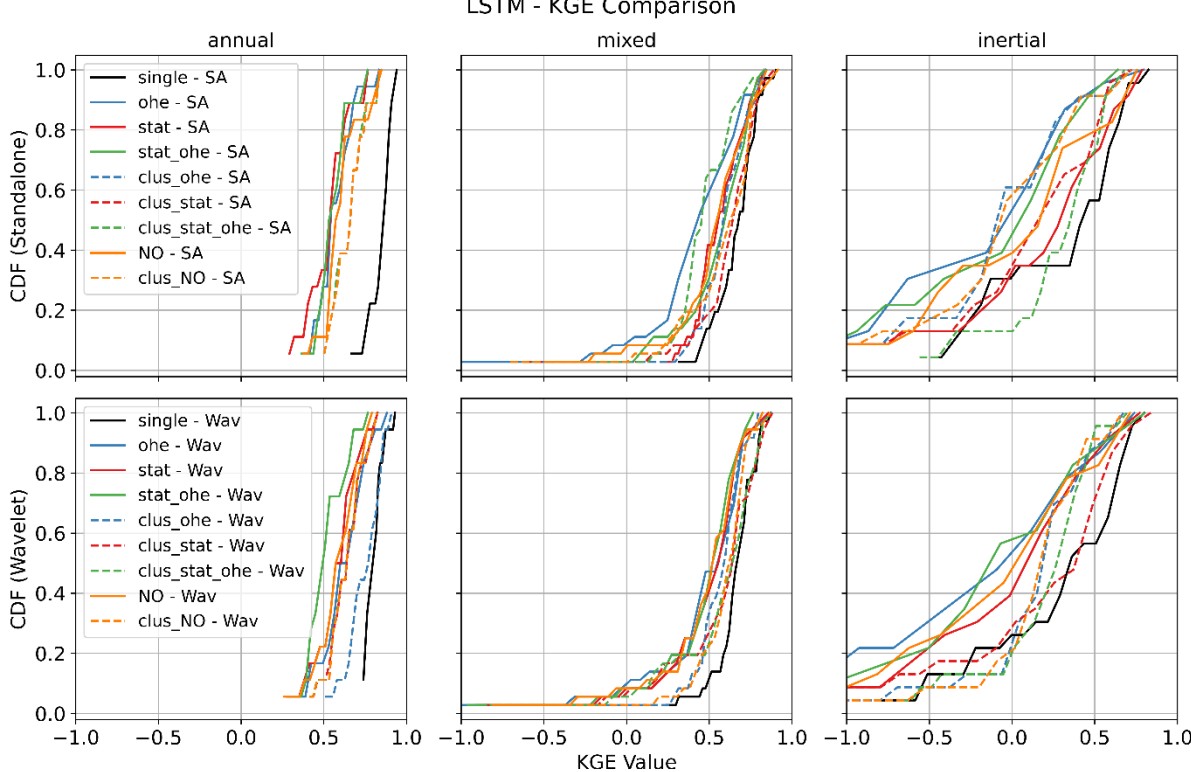

Figure A1: CDF Comparison of KGE values of the LSTM With different approaches and GWL types.

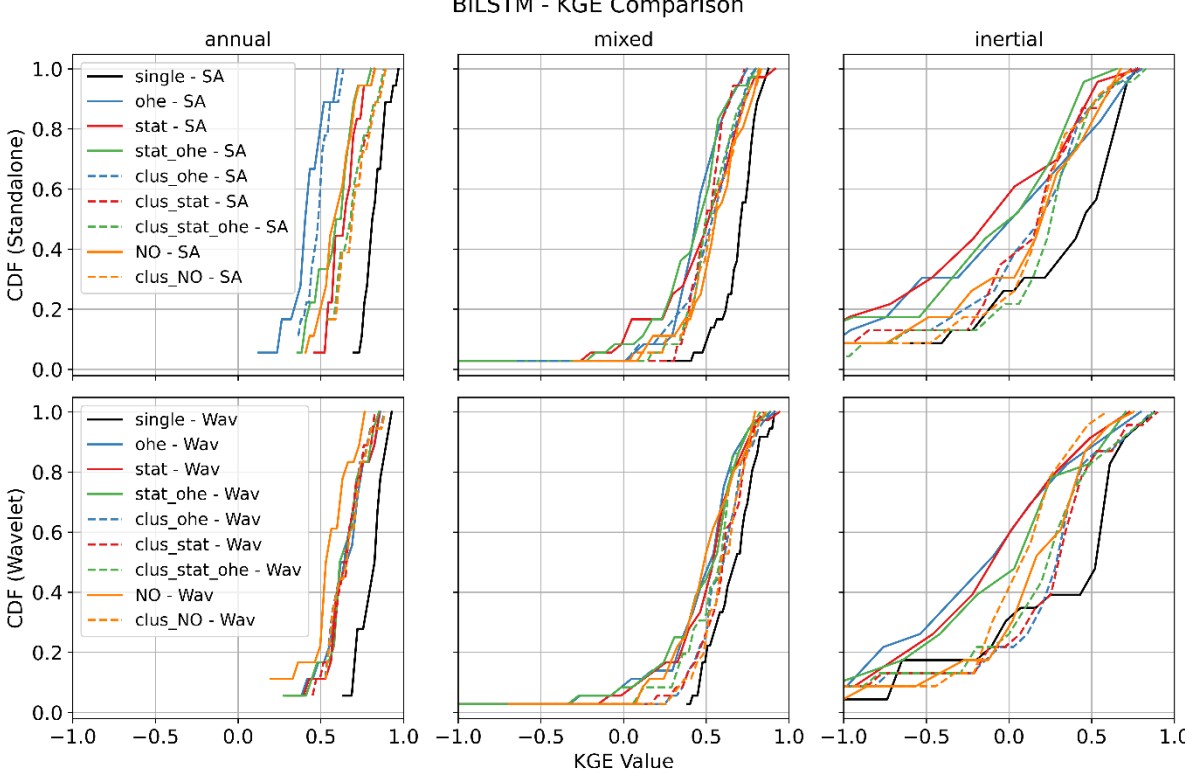

Figure A2: CDF Comparison of KGE values of the BiLSTM With different approaches and GWL types.

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
