# Peer review of "Training deep learning models with a multi-station approach and static aquifer attributes for groundwater level simulation: what's the best way to leverage regionalised information?"

_EGUsphere, 2024_

## Author Comment (AC1)

**Response to Reviewer 1 of the Manuscript: Training deep learning models with a multi-station approach and static aquifer attributes for groundwater level simulation: what's the best way to leverage regionalised information?**

**We appreciate your insightful comments and suggestions for improving our manuscript. Please find our point-by-point responses below in bold.**

*Chidepudi et al. used deep learning approaches to simulate and predict groundwater level dynamics. Authors compared and discussed the performance of different approaches of different combinations, such as different DL models, different inputs (i.e., dynamic factors and static factors), wavelet decomposition of precipitation, one-hot encoding etc. Using deep learning approach to simulate and predict dynamic groundwater levels is challenging. This work is important and could be a good reference for the community. The paper is generally well organized but there are still a lot of details unclear. Major revision is needed for further review.*

**Thank you for acknowledging the importance and potential impact of our work. We will address all the unclear details in the revised manuscript as described below.**

1. *There are no clear introductions of model structures.*

**We will enhance the details in the revised version. Specifically, we will include a new subsection in Section 3.1 (Theoretical modelling background) after line 205, providing details of the model structures, including the number of layers, units, and other relevant other common hyperparameter details (as shown in Table 1) for LSTM, GRU, and BiLSTM models. Though these models are widely used nowadays in all other subfields of hydrology as highlighted in line 224, if needed we will provide cell diagrams in supplementary document to explain the main principle of each network type.**

**Table 1: Hyperparameter details (Modified and adapted from chidepudi et.,al 2023)**

| Hyperparameter | Value considered |
| --- | --- |
| Sequence length | 48 |
| Dropout | 0.2 |
| Optimizer | ADAM |
| Early stopping | 50 |

| | |
|---|---|
| **Number of layers** | 1 |
| **Hidden neurons** | **(10, 20, …,100) by 10** |
| **Learning rate** | **(0.001,0.01) (log values)** |
| **Batch size** | **(16, 32, …,256) by powers of 2** |
| **Epoch** | **(50, 100, …,500)** |

2. *I didn't find details of the model input or the structure of the input data. I especially wanted to know this in the multi-station approach*

We will clarify the input data used for each of the multi-station approaches with standalone and wavelet assisted models by enhancing Section 3.2 (Experimental design) after line 240. This subsection will explain the structure and preprocessing of the input data, including the dynamic variables and static attributes, and how they were combined and formatted as input to the models. All the models use sequences as input for point simulation. The input data is structured as a 3D tensor with dimensions (samples, sequence_length, num_features), where the sequence length is set to 48 (4 years of monthly data), and the number of features includes both dynamic (time series of precipitation, temperature, surface net solar radiation…) and one-hot encoded static variables depending on the type of approach. For wavelet models, dynamic variables are also time series that are wavelet components of original inputs (time series of precipitation, temperature, surface net solar radiation…).

3. *How did you choose the training and test sets?*

In Section 3.2 (Experimental design), after line 295, we will add a new paragraph detailing the selection of training and test sets for the different modeling approaches. For the single-station approach, the data was split into training (80%) and testing sets (20%) as described in Chidepudi et al., 2023. Furthermore, to facilitate hyperparameter tuning, the last 20% of the training data was used as a validation set. For the multi-station approach, the train-test split was also performed at each station, following the same procedure as the single-station approach. However, the data from all stations was then collectively combined

during the training. The rationale behind the specific train-test split is to ensure that the models capture the multi-annual to decadal variability in groundwater levels (GWLs) observed in the region. To achieve this, a minimum of 34 years of data (1970-2014) was used for training, while the most recent 8.66 years of data (2015/01-2023/08) were reserved for testing. This split corresponds to approximately 80% of the data for training and 20% for testing. By following this approach, we aimed to ensure that the models were exposed to a sufficiently long period of data during training, enabling them to capture the amplitude and variability of GWL fluctuations over multi-annual to decadal timescales. The testing period was chosen to be the most recent years, allowing for an evaluation of the models' performance on the latest available data.

4. *I didn't find how large your research area (only a figure). The resolution of ERA5 is low and the true variations of these hydrometeorological variables may not be accurately presented by the products*
5. *What do you think about the uncertainties of data products from ERA5*

**Common response for both these comments (4&5) as they seem somehow related.**

**Regarding the research area, we will include additional details on research area in Section 2 (Study Area) to clearly specify the geographic extent covered in our study which is approximately 80,000 km2 covering two main basins (Seine and Somme).**

**In Section 2 (Data), after line 160, we will also discuss the implications of spatial resolution on capturing local variations when using data products like ERA5.**

**While we understand your concern about the potential limitations in accurately representing localized groundwater dynamics, ERA5 is the best available global reanalysis with the data available from 1940 and is generally considered adequate for capturing regional and global hydrometeorological variations. ERA5 Reanalysis data do have the uncertainty related to potential regional biases, this is ongoing debate as being discussed in (Maria Clerc-Schwarzenbach et al., 2024.) CAMELS (ERA5) vs CARAVAN (ERA5-Land) paper. Precipitation is considered to have more bias than temperature. However, for our study area, we have been evaluating different potential alternative reanalysis products, such as the Safran reanalysis developed specifically for France (Vidal et al., 2010). It appeared that both ERA 5 and Safran precipitation contained the same low-frequency components as detected in GWL time series as displayed in Fig.2 (this paper) and Fig.11 in Chidepudi et al 2023. ERA 5 then seems quite suitable for our purpose.**

**Discussing uncertainty of ERA5 is beyond the scope of this paper and can be considered research work as itself. However, we will add relevant references that discussed this point.**

*6. Did you only conduct the wavelet decomposition on precipitation or other variables also?*

**We will clarify that wavelet decomposition is done only on input dynamic variables after line 200: wavelet decomposition is being performed on time series only, each input time series being eventually replaced with its 5 wavelet components (corresponding to the decomposition level selected).**

*7. What is the resolution of the data products of static attributes?*

**In Section 2 (Data), after line 165, we will provide information on the resolution and sources of the static attribute data used**

**Static attributes are available for different ranges of aquifer classes with different resolutions, and we took the one that was associated with the Well IDs. Static attributes, coming from BDLISA database, are point-scale information, i.e., each well received set of attributes given different possible methods (geographical imputation, rule-based, human expertise). BDLISA is based on a mix of information coming from geological maps at a scale of 25km, piezometric maps, hydrochemistry, etc.**

**BDLISA was originally designed at a 25km scale and later upscaled to larger scales. For our study, we kept information coming from BDLISA at its original scale (25km), which means aquifer static attributes have a resolution of 25km. This should be understood as a local to regional description of aquifers.**

*8. What do you think the effects of hydraulic conductivity, elevation, slope etc. static attributes.*

**The decision to include the relevant static attributes comes from a trade-off between transposability of models and availability of attributes, as we have to make sure that all those variables are widely available at required resolution. Also, for some attribute like hydraulic conductivity, it might not be straightforward to get the most relevant resolution which is needed to account for the most appropriate characteristic describing the well. For instance, 25km resolution might not be relevant when aquifers are highly heterogeneous. Exploring the role of static attributes in more details would require much further works than what was conducted in this study.**

**9.** *Location of the well, i.e., in confined or unconfined aquifers may also be important*

All the wells considered in the study are in unconfined aquifers.

We would be happy to respond to any further comments while the discussion phase is still in progress.

**References**

Chidepudi, S. K. R., Massei, N., Jardani, A., Henriot, A., Allier, D., & Baulon, L. (2023). A wavelet-assisted deep learning approach for simulating groundwater levels affected by low-frequency variability. Science of the Total Environment, 865, 161035. https://doi.org/10.1016/j.scitotenv.2022.161035

Maria Clerc-Schwarzenbach, F., Selleri, G., Neri, M., Toth, E., van Meerveld, I., & Seibert, J. (2024). HESS Opinions: A few camels or a whole caravan? https://doi.org/10.5194/egusphere-2024-864

Vidal, J.P., Martin, E., Franchistéguy, L., Baillon, M., Soubeyroux, J.M., 2010. A 50-year high- resolution atmospheric reanalysis over France with the Safran system. Int. J. Climatol. 30 (11), 1627–1644. https://doi.org/10.1002/joc.2003

---

## Author Response (AR1)

**Dear Editor and Reviewers,**

**Thank you for your constructive feedback. We appreciate your insightful comments and suggestions for improving our manuscript. Please find our point-by-point responses below in bold.** Details of line numbers where changes were made are in red color. **Colors in some figures were updated following color-blind simulator as suggested by editorial team.**

**Response to Reviewer #1 of the Manuscript: Training deep learning models with a multi-station approach and static aquifer attributes for groundwater level simulation: what's the best way to leverage regionalised information?**

*Chidepudi et al. used deep learning approaches to simulate and predict groundwater level dynamics. Authors compared and discussed the performance of different approaches of different combinations, such as different DL models, different inputs (i.e., dynamic factors and static factors), wavelet decomposition of precipitation, one-hot encoding etc. Using deep learning approach to simulate and predict dynamic groundwater levels is challenging. This work is important and could be a good reference for the community. The paper is generally well organized but there are still a lot of details unclear. Major revision is needed for further review.*

**Thank you for acknowledging the importance and potential impact of our work. We addressed all the unclear details in the revised manuscript as described below.**

   1.  *There are no clear introductions of model structures.*

**We enhanced the details in the revised version. Specifically, we included a new subsection in Section 3.1 (Theoretical modelling background), providing details of the models, including the number of layers, units, and other relevant common hyperparameter details (as shown in Table 1) for LSTM, GRU, and BiLSTM models. Though these models are widely used nowadays in all other subfields of hydrology, as highlighted in line 261, if needed, we will provide cell diagrams in a supplementary document to explain the main principle of each network type. In addition, all the final hyperparameters are provided in the supplementary document.**

**Table 1: Hyperparameter details (Modified and adapted from chidepudi et.,al 2023)**

| Hyperparameter | Value considered |
| --- | --- |
| Sequence length | 48 |
| Dropout | 0.2 |
| Optimizer | ADAM |
| Early stopping | 50 |
| Number of layers | 1 |
| Hidden neurons | (10, 20, …,100) by 10 |
| Learning rate | (0.001,0.01) (log values) |
| Batch size | (16, 32, …,256) by powers of 2 |
| Epoch | (50, 100, …,500) |

Changes in lines 220-263 and Tables S3-S9 in the supplement.

2. *I didn't find details of the model input or the structure of the input data. I especially wanted to know this in the multi-station approach*

**We clarified the input data used for each multi-station approach with standalone and wavelet models by providing a figure with the number of covariates in Section 3.2 (Experimental design) after line 276 All the models use sequences as input for point simulation. The input data is structured as a 3D tensor with dimensions (samples, sequence_length, num_features) (Provided in Tables S5), where the sequence length is set to 48 (4 years of monthly data), and the number of features includes both dynamic (time series of precipitation, temperature, surface net solar radiation...) and one-hot encoded static variables depending on the type of approach. For wavelet models, dynamic variables are also time series that are wavelet components of original inputs (time series of precipitation, temperature, surface net solar radiation...).**

Figure 5 and Table S5

3. *How did you choose the training and test sets?*

In Section 3.2 (Experimental design), after line 345, we added a new paragraph detailing the selection of training and test sets for the different modelling approaches. "For the single-station approach, the data was split into training (80%) and testing sets (20%) as described in Chidepudi et al., 2023. Furthermore, the last 20% of the training data was used as a validation set to facilitate hyperparameter tuning. For the multi-station approach, the train-test split was also performed at each station, following the same procedure as the single-station approach. However, all station data was collectively combined during the training. The rationale behind the specific train-test split is to ensure that the models capture the multi-annual to decadal variability in groundwater levels (GWLs) observed in the region. To achieve this, a minimum of 34 years of data (1970-2014) was used for training, while the most recent 8.66 years of data (2015/01-2023/08) were reserved for testing. This split corresponds to approximately 80% of the data for training and 20% for testing. By following this approach, we aimed to ensure that the models were exposed to a sufficiently long period of data during training, enabling them to capture the amplitude and variability of GWL fluctuations over multi-annual to decadal timescales. The testing period was chosen to be the most recent years, allowing for an evaluation of the model's performance on the latest available data."

Lines 346-359 and Table S2

4. *I didn't find how large your research area (only a figure). The resolution of ERA5 is low and the true variations of these hydrometeorological variables may not be accurately presented by the products*
5. *What do you think about the uncertainties of data products from ERA5*

A common response for both these comments (4&5) as they seem somehow related.

Regarding the research area, we included additional details on the research area in Section 2 (Study Area) to clearly specify the geographic extent covered in our study which is approximately 80,000 km2 .

In Section 2 (Data), after line 160, we discussed the implications of spatial resolution on capturing local variations when using data products like ERA5.

While we understand your concern about the potential limitations in accurately representing localised groundwater dynamics, ERA5 is the best available global reanalysis with the data available from 1940. It is generally considered adequate for capturing regional and global hydrometeorological variations. ERA5 Reanalysis data do have uncertainty related to potential regional biases; this is an ongoing debate, as discussed in (Maria Clerc-Schwarzenbach et al., 2024). Precipitation is considered to have more bias than temperature. However, for our study area, we have been evaluating different potential alternative reanalysis products, such as the Safran reanalysis developed

specifically for France (Vidal et al., 2010). It appeared that both ERA 5 and Safran precipitation contained the same low-frequency components as detected in GWL time series as displayed in Fig.2 (this paper) and Fig.11 in Chidepudi et al 2023. ERA 5 then seems quite suitable for our purpose.

Discussing uncertainty of ERA5 is beyond the scope of this paper and can be considered research work as itself. However, we added relevant references that discussed this point.

Lines 145-185 , Figure 1

6. *Did you only conduct the wavelet decomposition on precipitation or other variables also?*

We clarified that wavelet decomposition is done only on input dynamic variables after line 200: wavelet decomposition is being performed on time series only, each input time series being eventually replaced with its 5 wavelet components (corresponding to the decomposition level selected).

Line 236-237

7. *What is the resolution of the data products of static attributes?*

In Section 2 (Data), after line 186, we provided information on the resolution and sources of the static attribute data used

Static attributes are available for different ranges of aquifer classes with different resolutions, and we took the one that was associated with the Well IDs. Static attributes, coming from BDLISA database, are point-scale information, i.e., each well received set of attributes given different possible methods (geographical imputation, rule-based, human expertise). BDLISA is based on a mix of information coming from geological maps at a scale of 25km, piezometric maps, hydrochemistry, etc.

BDLISA was originally designed at a 25km scale and later upscaled to larger scales. For our study, we kept information coming from BDLISA at its original scale (25km), which means aquifer static attributes have a resolution of 25km. This should be understood as a local to regional description of aquifers.

Lines 186-201, Table S1 and Figure 2

8. *What do you think the effects of hydraulic conductivity, elevation, slope etc. static attributes.*

The decision to include the relevant static attributes comes from a trade-off between the transposability of models and the availability of attributes, as we have to make sure that all those variables are widely available at the required resolution. Also, for some attributes like hydraulic conductivity, it might not be straightforward to get the most relevant resolution, which is needed to account for the most appropriate characteristic describing the well. For instance, 25km resolution might not be relevant when aquifers are highly heterogeneous.  Exploring the role of

**static attributes in more details would require much further works than what was conducted in this study.**

> **9.** *Location of the well, i.e., in confined or unconfined aquifers may also be important*

**All the wells considered in the study are in unconfined aquifers.**

**Response to Reviewer #2 of the Manuscript: Training deep learning models with a multi-station approach and static aquifer attributes for groundwater level simulation: what's the best way to leverage regionalised information?**

*Review comments on the manuscript: Training deep learning models with a multi-station approach and static aquifer attributes for groundwater level simulation: what's the best way to leverage regionalised information? by Chidepudi et al.*

*The manuscript presents several different deep learning approaches to simulate groundwater levels. Dynamic as well as static variables are used to train deep learning models to represent fluctuations on a high temporal resolution (daily data) in northern France. These different deep learning models were combined with different sets of input data (including preprocessing) and training strategies. Overall, the work is timely and covers the important topic of data-driven approaches to simulate dynamic groundwater levels. However, the manuscript has several shortcomings which are listed below. Major revision is needed.*

**Thank you for taking time to give detailed and constructive comments. We will address all the remaining issues listed in a revised version following our responses below.**

*Main Comments*

*What is the best way to leverage regionalised information? - The authors raise this question in the manuscript title but in my opinion, they do not answer the question in a sufficient way. This has mainly two reasons:*

- *The manuscript seems to be a combination of a technical note and a case study which leads to the result that a lot of essential information are missing. Reviewer 1 already pointed out several of the technical issues. In addition, a description of the data set is entirely missing. The only information available for the reader is the rough distance between the observation wells and the density in the region. Important information to understand the results and therefore the feasibility of the applied methodology is not supplied by the authors. For example: What is the distribution of static attributes in the different cluster groups? Looking at the attributes presented in Table 1, large differences between lithologies are to be expected (e.g karst vs. clay). Could it be that the annual group consist mainly of observation wells located in karstic/fractured areas and what would this tell us about the outcome of the study? Are these static attributes even presented/discussed in Chapter 4 (I assume that you can see them in Fig. 9 but they are not even named somewhere?*

**Technical issues also pointed out by Reviewer #1 mainly concerned the presentation of the hyperparameters eventually selected or optimized, and the architectures of the recurrent-based models. We explained how these comments can be addressed in our response to Reviewer #1. Regarding dataset presentation issue: in the version**

of the paper submitted, we presented the databases used, including the number duration, and sampling rate of the groundwater level time series, as well as a table of static attributes. Missing information or not provided at the right place, as pointed out by Reviewer #2 (e.g., the number of stations in each class, which was initially presented in the discussion section)  moved to the appropriate data section. For instance, we added an in-depth comparison of attributes available for different types of groundwater levels, along with improved details of the datasets. The three static attributes for different types of groundwater are shown in the pie plots below. However, it is important to keep in mind that such information is always very local and only valid for a given well. A full description of all these attributes  included in the form of a table in the supplement.

**Figure 2 and Tables S1 , Lines 145-215**

- *The presentation and discussion of the results lacks the already mentioned discussion of the regional context but also a discussion of the results in a broader context. For example, the authors write L398:"However, wavelet pre-processing shifts the importance towards dynamic components, reducing the contributions of static features or OHE. When clustering is combined with wavelet preprocessing, low-frequency precipitation components emerge as key contributors, improving model performance.*

   *Does this mean that the importance of all dynamic components is higher by default, and we do not need to consider geological/hydrodynamic/topographic features? Does this apply to all kind of unconfined aquifer systems (shallow, deep, karstic…)? Here it would be interesting to combine/compare your results with/to other available publications considering static attributes on a regional scale (e.g. Heudorfer et al., 2024 or Haaf et al., 2023).*

   **In the present paper, we aimed only to assess whether, in our context of relatively parsimonious availability of basin properties, considering such attributes within the framework of DL modeling would significantly improve the simulations. For the sake of the generalization capabilities of DL models, we also probably need to find a reasonable trade-off between the use of all possible/relevant static features and their availability over large areas.**

   **We cannot expect static characteristics to be more important than precipitation, temperature, or other variable time series of the water cycle in explaining groundwater level (GWL) time series variations. As mentioned above, the aim here is to assess to what extent available static attributes, in combination with indispensable forcing hydrological variables, may help refine and improve GWL simulations for stations in various (hydro)geological contexts. This (hydro)geological information is largely accounted for in the weights of the neural network model, but the question remains whether additional static information can be helpful. Our results suggest that in some cases, particularly for the most inertial groundwater level types that mainly record low-frequency, climate-like information, improvements can be gained by adding static features.**

   **We agree that a more thorough comparison with papers that have used static attributes on a regional scale was needed and now added to the discussion section.**

   **Since the purpose of the paper presented here is not to determine the forcing factors of groundwater level variations, comparison with such state-of-the-art studies will help to put our results into perspective, inasmuch as a comprehensive evaluation of such links would require specific approaches. Such approaches have already been undertaken and presented in numerous previous works that we will use to feed the discussion about this important topic as in (Lee et al., 2019; Heudorfer et al., 2019; Liesch and Wunsch, 2019; Haaf et al., 2020; Giese et al., 2020). In our own previous works (albeit for the Normandy region only), the linkages between groundwater level variability and potential forcing factors such as the thickness and lithology of surficial**

**formations, aquifer thickness, vadose zone thickness, upstream/downstream location along the flow path, distance to the river, presence of karst, etc., were investigated using dedicated approaches (Slimani et al., 2009; El Janyani et al., 2012, 2014).**

Lines 516-543, 576-595

*The quality in writing (language, clarity etc.) differs a lot throughout the manuscript. This makes it difficult to follow the central theme and therefore requires revision. Sometimes sentences reoccur, e.g. L73: DL models have proved effective on a local scale, and are also on a larger scale by collectively training a significant number of piezometers (Chidepudi et al., 2023b; Heudorfer et al., 2024) vs. L80: The DL models have proved effective at local scale and are also proving more effective on a larger scale. At the same time the introduction of terms and abbreviation is totally off, some examples: GWLs is first introduces in the Introduction and then again in line 185, 308, 378 and 436; SHAP is first introduced in line 231 and then again in 461; an introduction (even though they are quiet common) for AI/DL/KGE and NSE is entirely missing. Altogether it feels like sections/paragraphs of different origin were put together.*

**We improved the text with appropriate introduction of terms wherever needed. Also the entire text checked for homogenization of the writing quality.**

All over the text

*Secondary Comments*

*L85: sensitivity to human activities - I do not really understand why this is an **additional challenge compared to runoff data**. Does it mean runoff data are not sensitive to human activities (e.g. river straightening, dam construction etc.)?*

**We agree that "additional challenge" was certainly not the most appropriate term. Here we meant to say that groundwater level data are affected by different types of challenges with respect to human activities. This can be confusing and then modified in the text.**

Lines 74-79

*L121: their application to GWL simulation is still questionable. – Do you really mean questionable?*

**We agree "questionable" is clearly not the right term.  revision: "their application to GWL simulation is still not fully explored or validated across diverse hydrogeological settings."**

Line 108

*L141: We refer to (Beven and Young, 2013), for differences in the use of the terms simulation and forecasting. - I do not see the connection between the sentence and the rest of the paragraph. Maybe a few more words are needed?*

**We updated it as : "We would like to highlight at this point that the present study is not dedicated to 'forecasting' as it is the case in most applications of DL to groundwater modeling. The reader can be referred to Beven and Young (2013) for distinctions between 'simulation' and 'forecasting'. In brief, according to their framework, 'simulation' means reproducing system behavior without using observed outputs, while 'forecasting' involves reproducing system behavior ahead of time based on past observations. This study focuses on simulation to understand GWL dynamics, rather than forecasting future levels. This distinction is important for framing our approach and interpreting our results."**

Lines 131-136

*L164: Although they seem somehow redundant, they are expected to provide complimentary information about the hydrogeological nature of the hydrosystems – This could and should be tested at one point (which does not mean that you have to add it here).*

**We agree that it would certainly be interesting to conduct some statistical analysis (multivariate, for instance) to assess the potential redundancy of the information provided by the different static features, but 1- we agree with reviewer #2 that this should probably be undertaken in the framework of one dedicated study (cf. our response to some previous comments), 2) from the DL point of view, redundancy should not be an issue, DL models are basically designed to handle (and learn from) as much information as possible without taking into account any possible redundancy within the data (the model will adjust its parameters according to the most useful information detected). For instance, one part of the useful information can be common to 2 features, and at the same time one other part can be specific to each. It will not be detrimental to the performance of the model. As hydrologists, we only ensured that the input data are hydro-geologically relevant (albeit strictly speaking, from the DL standpoint, the models can even get rid of irrelevant data itself during the learning process).**

*L167/ L173/180/323: Baulon et al., 2022a/b?*

**Corrected to a/b in all instances.**

*L187: Bidirectional LSTM - I would be good to provide a reference especially since you write in L192: BiLSTM […] are particularly good at identifying various patterns in data sequences, making them ideal for simulating GWLs that change over time. or is this already a result of your study?*

This was not the outcome of this study but a general advantage of the model and references will be provided.

Lines 222-229

*L304: Further explanation needed. The figure does not provide any details, especially no comparison, as written by the authors.*

**We agree these 2 sentences are confusing. It is also true that the difference between the various models is never extremely noticeable, because all the models performed well eventually. A thorough examination of the results of figure 3 (comparison of the 3 model types in single-station mode) and of figure 5 with figures A1 and A2 led us to the conclusion that GRU performed slightly better. Another reason why GRU was preferred is also related to its computational efficiency. Since the difference in performances is not very noticeable, we suggest the following modification:**

**"All models tested in the case of this study, performed more or less equivalently and eventually led to very satisfactory results. This can be attested by performance comparison shown in figure 3 (comparison of the 3 model types in single-station mode) and by comparing figure 5 with figures A1 and A2 (multi-station mode). We finally decided to favor the GRU architecture owing to its recognised computational efficiency over more traditional LSTM-based architectures (Cho et al., 2014; Cai et al., 2021; Chidepudi et al., 2023, 2024 )".**

Lines 369-374

*L355: This is an information you expect earlier in the manuscript.*

**Agreed, we moved this information to the data section for better context.**

Line 155

*L372: Why do you formulate "new research questions" here, is this necessary?*

**We agree, formulating new research questions again at this stage can be misleading. We then removed them as it doesn't change the discussion.**

*L425: No_ohe_no_stat approach?*

**We updated it to use consistent and clear naming conventions for all approaches throughout the paper.**

Line 512, Figures 12 & 13

*References: Nourani, V., Alami, M. T., & Vousoughi, F. D. (2015).  - I do not find a citation of this paper.*

**Corrected the citation in** line 101

*References:*

- *Heudorfer, B., Liesch, T., & Broda, S. (2024). On the challenges of global entity-aware deep learning models for groundwater level prediction. Hydrol. Earth Syst. Sci, 28, 525–543. https://doi.org/10.5194/hess-28-525-2024*
- *Haaf, E., Giese, M., Reimann, T., & Barthel, R. (2023). Data-driven estimation of groundwater level time-series at unmonitored sites using comparative regional analysis. Water Resources Research, 59, e2022WR033470. https://doi.org/10.1029/2022WR033470*
- **Slimani, S., Massei, N., Mesquita, J. et al. Combined climatic and geological forcings on the spatio-temporal variability of piezometric levels in the chalk aquifer of Upper Normandy (France) at pluridecennal scale. Hydrogeol J 17, 1823–1832 (2009). https://doi.org/10.1007/s10040-009-0488-1**
- **El Janyani, S., Dupont, JP., Massei, N. et al. Hydrological role of karst in the Chalk aquifer of Upper Normandy, France. Hydrogeol J 22, 663–677 (2014). https://doi.org/10.1007/s10040-013-1083-z**
- **Sanae El Janyani, Nicolas Massei, Jean-Paul Dupont, Matthieu Fournier, Nathalie Dörfliger. Hydrological responses of the chalk aquifer to the regional climatic signal, Journal of Hydrology,Volumes 464–465,2012, Pages 485-493,ISSN 0022-1694, https://doi.org/10.1016/j.jhydrol.2012.07.040**
- **Giese, M., Haaf, E., Heudorfer, B., & Barthel, R. (2020). Comparative hydrogeology – reference analysis of groundwater dynamics from neighbouring observation wells. Hydrological Sciences Journal, 65(10), 1685–1706. https://doi.org/10.1080/02626667.2020.1762888**
- **Haaf, E., Giese, M., Heudorfer, B., Stahl, K., & Barthel, R. (2020). Physiographic and climatic controls on regional groundwater dynamics. Water Resources Research, 56, e2019WR026545. https://doi.org/10.1029/2019WR026545**
- **Heudorfer, B., Haaf, E., Stahl, K., & Barthel, R. (2019). Index-based characterization and quantification of groundwater dynamics. Water Resources Research, 55, 5575–5592. https://doi.org/10.1029/2018WR024418**
- **Lee, S., Lee, KK. & Yoon, H. Using artificial neural network models for groundwater level forecasting and assessment of the relative impacts of influencing factors. Hydrogeol J 27, 567–579 (2019). https://doi.org/10.1007/s10040-018-1866-3**
- **Tanja Liesch, Andreas Wunsch, Aquifer responses to long-term climatic periodicities,Journal of Hydrology,Volume 572,2019,Pages 226-242,ISSN 0022-1694,https://doi.org/10.1016/j.jhydrol.2019.02.060**
- **Hejiang Cai, Haiyun Shi, Suning Liu, Vladan Babovic, Impacts of regional characteristics on improving the accuracy of groundwater level prediction using machine learning: The case of central eastern continental United States,Journal of Hydrology: Regional Studies, Volume 37,2021,100930,ISSN 2214-5818, https://doi.org/10.1016/j.ejrh.2021.100930**

- Cho, K., Van Merriënboer, B., Bahdanau, D., & Bengio, Y. (2014). On the properties of neural machine translation: Encoder-decoder approaches. arXiv preprint arXiv:1409.1259

-  Maria Clerc-Schwarzenbach, F., Selleri, G., Neri, M., Toth, E., van Meerveld, I., & Seibert, J. (2024). HESS Opinions: A few camels or a whole caravan? https://doi.org/10.5194/egusphere-2024-864

-  Vidal, J.P., Martin, E., Franchistéguy, L., Baillon, M., Soubeyroux, J.M., 2010. A 50-year high- resolution atmospheric reanalysis over France with the Safran system. Int. J. Climatol. 30 (11), 1627–1644. https://doi.org/10.1002/joc.2003

- Chidepudi, S. K. R., Massei, N., Jardani, A., Henriot, A., Allier, D., & Baulon, L. (2023). A wavelet-assisted deep learning approach for simulating groundwater levels affected by low-frequency variability. Science of the Total Environment, 865, 161035. https://doi.org/10.1016/j.scitotenv.2022.161035

---

## Author Response (AR2)

**Dear Editor,**

**Thank you for your constructive feedback. We appreciate your insightful comments and feedback for improving our manuscript. Please find our point-by-point responses below in bold; the original comments are in italics.**

*The authors have considered the main critical issues raised by the reviewers during the revision process. For this reason, while in my previous assessment I was inclined in requiring a further revision process, after a thorough analysis of the manuscript I decided not to go for a second round of review.*

*This been said, upon a careful review of the manuscript, I must highlight that it still contains a significant number of typos and awkwardly constructed sentences that require the attention of the authors. While I understand the time constraints, especially with one of the authors being a Ph.D. student nearing graduation, it is crucial to uphold the quality of the manuscript for the benefit of both the journal and yourselves for the purpose of effectively communicating your research. Below, I have highlighted few (non-exhaustive) examples of suggested changes to be made.*

**Thank you once again for your valuable feedback and for allowing us to enhance the quality of our manuscript. We have now addressed all the suggested corrections as below.**

*Line 25 " …to learn the dominant station …. preferentially…", I suggest to rephrase the sentence as "…to preferentially learn…"*
**Updated**

*Line 37 "…for grasping a more global view of water reserves.." Here the adjective "more" is not appropriate. I suggest rewriting as "for grasping a global view of water reserves"*
**Updated**

*Line 43 "Building the large -scale model…..." should be "Building a large -scale model…"*
**Updated**

*Line 46 "However, the numerical, physics-based representation of all the physical processes occurring during the hydrological cycle ….." The sentence is awkward; consider rewriting it, for example as "However, the numerical, physics-based, representation of all processes occurring during the hydrological cycle ….*
**Updated**

*Lines 160-161 and lines 168. The repetition of the sentence "All the wells considered in the study are in unconfined aquifers" should be rectified.*
**Corrected**

*Line 188 "Gualtieri (2022) highlighted that ERA5 uncertainties were greater in mountainous and particularly in coastal locations …..". Consider rewriting this sentence, for example as: "Gualtieri (2022) highlighted that ERA5 uncertainties are greatest at mountain and coastal locations …..*
**Updated**

*Line 193 "However, for our study area, we have been evaluating different potential alternative reanalysis products, such as….". Consider rewriting this sentence, for example as "For our study area, we evaluated several potential alternative reanalysis products, such as ….."*
**Updated**

*Line 205 "BDLISA was originally designed at a 25km scale and later upscaled to larger scales. For our study, we kept information coming from BDLISA at its original scale (25km), which means aquifer static attributes have a resolution of 25km".*
*Since you are using the original scale (without upscaling), I would remove the sentence "For our study…", to avoid confusing the reader.*
**Removed**

*Line 235 "… we have to make sure" replace by "…we need to ensure…"*
**Updated**

*Page 9. Remove the Figure without number and caption, it is already included in Fig. 1.*
**This figure was only visible (with striked out) in previous tracked version only to represent the change but this is now avoided in the current revision**

*Line 238 "Also, for some attributes like hydraulic conductivity, it might not be straightforward to get the most relevant resolution, which is needed to account for the most appropriate characteristic describing the well…". Please consider rephrasing this sentence, did you mean "describing the aquifer"?*
**Updated**
*Line 272 "Details of range of hyperparameters used are shown in Table 1". The correct reference should be Table 2*
**Updated**

*Line 379 "….Furthermore, to facilitate hyperparameter tuning, the last 20% of the training data was used as a validation set" and line 385 "This split corresponds to approximately 80% of 385 the data for training and 20% for testing." These two sentences convey the same information. Please rewrite.*

**Updated**

*Line 445 – 446.Clarify the basis for stating "While single station models perform best" as it is not evident from Fig. 8-9*

**Figures 8 to 10 show the best GWL simulations obtained of different types (annual, mixed and inertial) for single and multi-station models. For those particular cases, both approaches perform similarly and lead to good performance. However, the single-station seems to perform best for inertial GWL type for training by simple visual assessment, and it is clear from the comparison of KGE values of all stations (Fig.7) that the more specialised single-station models generally gave the best results overall, although not significantly. This is more specifically true for inertial GWL, where regional model performances reach the same level as single-station models. While single-station models perform well, multi-station models are valuable when single-station modelling is impractical due to data limitations or computational requirements. For instance, for inertial types where the length of training data might be an issue (e.g. Chidepudi et al., 2024), the performance of the wavelet multi-station models was completely comparable to single-station models (Fig.7, wavelet models/inertial types), showing that in the case of data limitation, the regional approach seems to compensate the lack of temporal depth of available time series.**

**Updated**

*Line 470. "In particular …".Consider rephrasing this sentence.*

**Updated as follows:**
**"The distribution of data points on the SHAP diagram indicates either a positive (right side on the x-axis) or negative (left side on the x-axis) impact on the output variable. In contrast, the colour scale indicates the range of feature values in which red represents large values, and blue represents small ones of the corresponding feature. Features (input variables) are organised from the most to the least influencing, from top to bottom, based on each feature's mean absolute SHAP values. For instance, in Figure 11a, total precipitation (tp) is the most**

**influencing feature on the GWL output, and the large feature values on the right (red) correspond to a positive influence on GWL (high GWL with high total precipitation). On the left-side, negative tp SHAP values indicate lower precipitation values contributing to the low GWLs."**

*Lines 552-560. The following sentence is unclear "In the framework of our study, we decided to exclude some relevant characteristics such as vadose or saturated zone thickness: even when averaged over quite long periods (several years), these values actually represent GWL (the target variable)….. These authors concluded that the models did not show any entity awareness and eventually utilized static attributes as simple identifiers (almost similar to the OHE approach presented herein), meaning that the models did not make use of the relevant (hydro)geological information." Please rewrite.*

**Updated as :**
**In the framework of our study, we decided to exclude characteristics such as vadose or saturated zone thickness. Such variables have been used in previous studies (El Janyani et al., 2012 and 2014; Haaf et al., 2023) and considered static (averaged over long periods of time) to investigate the impact of (hydro)geological and geomorphologic characteristics on GWL behaviours. Yet, in our study, it was not relevant to consider such characteristics as "static" since they are linked to the varying GWL which we aim to simulate. Other types of static characteristics reflecting the hydraulic properties of the aquifers, such as hydraulic conductivity, transmissivity, porosity or storativity, were also discarded. While informative in terms of hydrological knowledge, it is likely that: 1- their availability may not be guaranteed over large areas, hence limiting their usefulness. 2- their representativeness as numeric values might be questionable in contexts where spatial heterogeneity is high: in such cases, more general qualitative descriptors such as "fissured" or "porous" might be preferable, as using precise values of hydraulic conductivity, etc., would likely make the models very sensitive to hydraulic heterogeneity which can not be accounted for so precisely. In addition, in a recent and relevant study on "entity-aware deep learning models with static attributes," Heudorfer et al. (2024) highlighted that the models developed did not actually show any entity awareness and eventually utilised static attributes as simple identifiers (almost similar to the OHE approach presented herein), meaning that the models did not make use of relevant and precise (hydro)geological information.**

*Lines 542-545 "… for alluvial aquifers with probably quite high hydraulic conductivity overall" Clarify the meaning of "probably" here.*
**Removed the word "Probably" and indicated the median value from the study we were referring to.**

*Conclusions, line 595-600 "In this article, we introduced the following question: "What's the best way to leverage regionalised information?". In light of our results, it then seems like this is highly dependent on the amount and types of static attributes. It is generally expected that a much higher number of static attribute types would allow for a much better improvement of the multi-station simulation approach. However, Heudorfer et al. (2024) found no improvements using around 28 static features (including 18 environmental and ten time series-based). Also, as pointed out by these authors, employing static attributes for model training might be more relevant in applications on larger scales and/or larger datasets". Please consider rewriting these sentences. As written, it seems that your results are in contrast with the findings of Heudorfer et al. (2024). On the other hand, my understanding is that static attributes can be used for model training on large scales, while they are not particularly useful on small-moderate scales (as such investigated by Heudorfer et al., 2024). Please also include the magnitude levels for both "large" and "small" (or moderate scales.*

**Our conclusion does not contradict Heudorfer, and this is now clarified in the main text: we found that the most decisive component of improvement was Wavelet and clustering, and then static provides minor additional improv, which can be valuable but, again, not the most decisive.**

**Updated**

*Table 4. Fix the typo ("Lattitude")*
**Corrected**

*Additionally, I recommend enhancing the quality of Figures 1-2,8-13, several labels are blurred or not visible.*

**The figures have now been updated with increased sizes of all fonts and image resolutions.**